# Consistent Non-Parametric Methods for Maximizing Robustness

**Robi Bhattacharjee**
University of California San Diego
rcbhatta@eng.ucsd.edu

**Kamalika Chaudhuri**
University of California San Diego
kamalika@eng.ucsd.edu

## Abstract

Learning classifiers that are robust to adversarial examples has received a great deal of recent attention. A major drawback of the standard robust learning framework is there is an artificial robustness radius $r$ that applies to all inputs. This ignores the fact that data may be highly heterogeneous, in which case it is plausible that robustness regions should be larger in some regions of data, and smaller in others. In this paper, we address this limitation by proposing a new limit classifier, called the neighborhood optimal classifier, that extends the Bayes optimal classifier outside its support by using the label of the closest in-support point. We then argue that this classifier maximizes the size of its robustness regions subject to the constraint of having accuracy equal to the Bayes optimal. We then present sufficient conditions under which general non-parametric methods that can be represented as weight functions converge towards this limit, and show that both nearest neighbors and kernel classifiers satisfy them under certain conditions.

## 1 Introduction

Adversarially robust classification, that has been of much recent interest, is typically formulated as follows. We are given data drawn from an underlying distribution $D$, a metric $d$, as well as a pre-specified robustness radius $r$. We say that a classifier $c$ is $r$-robust at an input $x$ if it predicts the same label on a ball of radius $r$ around $x$. Our goal in robust classification is to find a classifier $c$ that maximizes astuteness, which is defined as accuracy on those examples where $c$ is also $r$-robust.

While this formulation has inspired a great deal of recent work, both theoretical and empirical [5, 17, 19, 20, 26, 15, 18, 21, 22, 23, 30], a major limitation is that enforcing a pre-specified robustness radius $r$ may lead to sub-optimal accuracy *and* robustness. To see this, consider what would be an ideally robust classifier the example in Figure 1. For simplicity, suppose that we know the data distribution. In this case, a classifier that has an uniformly large robustness radius $r$ will misclassify some points from the blue cluster on the left, leading to lower accuracy. This is illustrated in panel (a), in which large robustness radius leads to intersecting robustness regions. On the other hand, in panel (b), the blue cluster on the right is highly separated from the red cluster, and could be accurately classified with a high margin. But this will not happen if the robustness radius is set small enough to avoid the problems posed in panel (a). Thus, enforcing a fixed robustness radius that applies to the entire dataset may lead to lower accuracy and lower robustness.

In this work, we propose an alternative formulation of robust classification that ensures that in the large sample limit, there is no robustness-accuracy trade off, and that regions of space with higher separation are classified more robustly. An extra advantage is that our formulation is achievable by existing methods. In particular, we show that two very common non-parametric algorithms – nearest neighbors and kernel classifiers – achieve these properties in the large sample limit.

Our formulation is built on the notion of a new large-sample limit. In the standard statistical learning framework, the large-sample ideal is the Bayes optimal classifier that maximizes accuracy on the

35th Conference on Neural Information Processing Systems (NeurIPS 2021).

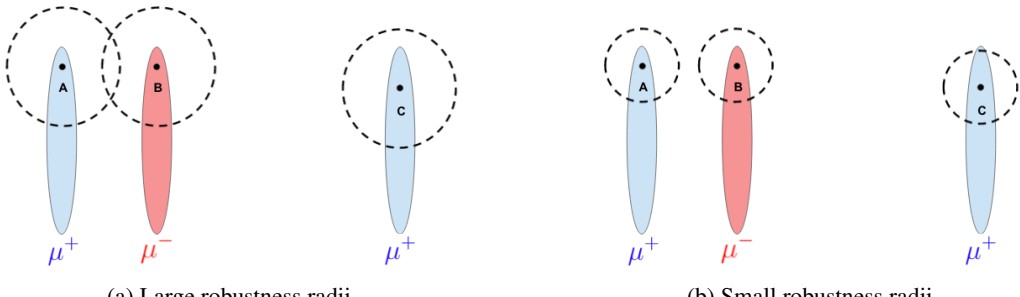

(a) Large robustness radii       (b) Small robustness radii

Figure 1: A data distribution demonstrating the difficulties with fixed radius balls for robustness regions. The red represents negatively labeled points, and the blue positive. If the robustness radius is set too large (panel (a)), then the regions of A and B intersect leading to a loss of accuracy. If the radius is set too small (panel (b)), this leads to a loss of robustness at point C where in principle it should be possible to defend against a larger amount of adversarial attacks.

data distribution, and is undefined outside. Since this is not always robust with radius $r$, prior work introduces the notion of an $r$-optimal classifier [31] that maximizes accuracy on points where it is also $r$-robust. However, this classifier also suffers from the same challenges as the example in Figure 1.

We depart from both by introducing a new limit that we call the neighborhood preserving Bayes optimal classifier, described as follows. Given an input $x$ that lies in the support of the data distribution $D$, it predicts the same label as the Bayes optimal. On an $x$ outside the support, it outputs the prediction of the Bayes Optimal on the nearest neighbor of $x$ *within* the support of $D$. The first property ensures that there is no loss of accuracy – since it always agrees with the Bayes Optimal within the data distribution. The second ensures higher robustness in regions that are better separated. Our goal is now to design classifiers that converge to the neighborhood preserving Bayes optimal in the large sample limit; this ensures that with enough data, the classifier will have accuracy approaching that of the Bayes optimal, as well as higher robustness where possible without sacrificing accuracy.

We next investigate how to design classifiers with this convergence property. Our starting point is classical statistical theory [25] that shows that a class of methods known as weight functions will converge to a Bayes optimal in the large sample limit provided certain conditions hold; these include $k$-nearest neighbors under certain conditions on $k$ and $n$, certain kinds of decision trees as well as kernel classifiers. Through an analysis of weight functions, we next establish precise conditions under which they converge to the neighborhood preserving Bayes optimal in the large sample limit. As expected, these are stronger than standard convergence to the Bayes optimal. In the large sample limit, we show that $k_n$-nearest neighbors converge to the neighborhood preserving Bayes optimal provided $k_n = \omega(\log n)$, and kernel classifiers converge to the neighborhood preserving Bayes optimal provided certain technical conditions (such as the bandwidth shrinking sufficiently slowly). By contrast, certain types of histograms do not converge to the neighborhood preserving Bayes optimal, even if they do converge to the Bayes optimal. We round these off with a lower bound that shows that for nearest neighbor, the condition that $k_n = \omega(\log n)$ is tight. In particular, for $k_n = O(\log n)$, there exist distributions for which $k_n$-nearest neighbors provably fails to converge towards the neighborhood preserving Bayes optimal (despite converging towards the standard Bayes optimal).

In summary, the contributions of the paper are as follows. First, we propose a new large sample limit the neighborhood preserving Bayes optimal and a new formulation for robust classification. We then establish conditions under which weight functions, a class of non-parametric methods, converge to the neighborhood preserving Bayes optimal in the large sample limit. Using these conditions, we show that $k_n$-nearest neighbors satisfy these conditions when $k_n = \omega(\log n)$, and kernel classifiers satisfy these conditions provided the kernel function $K$ has faster than polynomial decay, and the bandwidth parameter $h_n$ decreases sufficiently slowly.

To complement these results, we also include negative examples of non-parametric classifiers that do not converge. We provide an example where histograms do not converge to the neighborhood preserving Bayes optimal with increasing $n$. We also show a lower bound for nearest neighbors, indi-

cating that $k_n = \omega(\log n)$ is both necessary and sufficient for convergence towards the neighborhood preserving Bayes optimal.

Our results indicate that the neighborhood preserving Bayes optimal formulation shows promise and has some interesting theoretical properties. We leave open the question of coming up with other alternative formulations that can better balance both robustness and accuracy for all kinds of data distributions, as well as are achievable algorithmically. We believe that addressing this would greatly help address the challenges in adversarial robustness.

## 2 Preliminaries

We consider binary classification over $\mathbb{R}^d \times \{\pm 1\}$, and let $\rho$ denote any distance metric on $\mathbb{R}^d$. We let $\mu$ denote the measure over $\mathbb{R}^d$ corresponding to the probability distribution over which instances $x \in \mathbb{R}^d$ are drawn. Each instance $x$ is then labeled as $+1$ with probability $\eta(x)$ and $-1$ with probability $1 - \eta(x)$. Together, $\mu$ and $\eta$ comprise our data distribution $\mathcal{D} = (\mu, \eta)$ over $\mathbb{R}^d \times \{\pm 1\}$.

For comparison to the robust case, for a classifier $f : \mathbb{R}^d \to \{\pm 1\}$ and a distribution $\mathcal{D}$ over $\mathbb{R}^d \times \{\pm 1\}$, it will be instructive to consider its **accuracy**, denoted $A(f, \mathcal{D})$, which is defined as the fraction of examples from $\mathcal{D}$ that $f$ labels correctly. Accuracy is maximized by the **Bayes Optimal classifier**: which we denote by $g$. It can be shown that for any $x \in supp(\mu)$, $g(x) = 1$ if $\eta(x) \geq \frac{1}{2}$, and $g(x) = -1$ otherwise.

Our goal is to build classifiers $\mathbb{R}^d \to \{\pm 1\}$ that are both accurate and robust to small perturbations. For any example $x$, perturbations to it are constrained to taking place in the **robustness region** of $x$, denoted $U_x$. We will let $\mathcal{U} = \{U_x : x \in \mathbb{R}^d\}$ denote the collections of all robustness regions.

We say that a classifier $f : \mathbb{R}^d \to \{\pm 1\}$ is **robust** at $x$ if for all $x' \in U_x$, $f(x') = f(x)$. Combining robustness and accuracy, we say that classifier is **astute** at a point $x$ if it is both accurate and robust. Formally, we have the following definition.

**Definition 1.** *A classifier $f : \mathbb{R}^d \to \{\pm 1\}$ is said to be **astute** at $(x, y)$ with respect to robustness collection $\mathcal{U}$ if $f(x) = y$ and $f$ is robust at $x$ with respect to $\mathcal{U}$. If $\mathcal{D}$ is a data distribution over $\mathbb{R}^d \times \{\pm 1\}$, the **astuteness** of $f$ over $\mathcal{D}$ with respect to $\mathcal{U}$, denoted $A_{\mathcal{U}}(f, \mathcal{D})$, is the fraction of examples $(x, y) \sim \mathcal{D}$ for which $f$ is astute at $(x, y)$ with respect to $\mathcal{U}$. Thus*

$$A_{\mathcal{U}}(f, \mathcal{D}) = P_{(x,y) \sim \mathcal{D}}[f(x') = y, \forall x' \in \mathcal{U}_x].$$

**Non-parametric Classifiers** We now briefly review several kinds of non-parametric classifiers that we will consider throughout this paper. We begin with *weight functions*, which are a general class of non-parametric algorithms that encompass many classic algorithms, including nearest neighbors and kernel classifiers.

**Weight functions** are built from training sets, $S = \{(x_1, y_1), (x_2, y_2,), \ldots, (x_n, y_n)\}$ by assigning a function $w_i^S : \mathbb{R}^d \to [0, 1]$ that essentially scores how relevant the training point $(x_i, y_i)$ is to the example being classified. The functions $w_i^S$ are allowed to depend on $x_1, \ldots, x_n$ but must be independent of the labels $y_1, \ldots, y_n$. Given these functions, a point $x$ is classified by just checking whether $\sum y_i w_i^S(x) \geq 0$ or not. If it is nonnegative, we output $+1$ and otherwise $-1$. A complete description of weight functions is included in the appendix.

Next, we enumerate several common Non-parametric classifiers that can be construed as weight functions. Details can be found in the appendix.

**Histogram classifiers** partition the domain $\mathbb{R}^d$ into cells recursively by splitting cells that contain a sufficiently large number of points $x_i$. This corresponds to a weight function in which $w_i^S(x) = \frac{1}{k_x}$ if $x_i$ is in the same cell as $x$, where $k_x$ denotes the number of points in the cell containing $x$.

$k_n$-**nearest neighbors** corresponds to a weight function in which $w_i^S(x) = \frac{1}{k_n}$ if $x_i$ is one of the $k_n$ nearest neighbors of $x$, and $w_i^S(x) = 0$ otherwise.

**Kernel-Similarity classifiers** are weight functions built from a kernel function $K : \mathbb{R}_{\geq 0} \to \mathbb{R}_{\geq 0}$ and a window size $(h_n)_1^\infty$ such that $w_i^S(x) \propto K(\rho(x, x_i)/h_n)$ (we normalize by dividing by $\sum_1^n K((\rho(x, x_i)/h_n)))$.

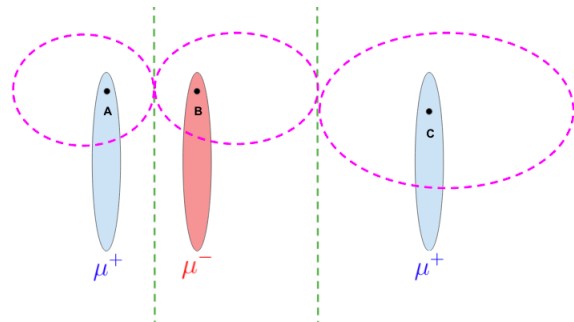

Figure 2: The decision boundary of the neighborhood preserving Bayes optimal classifier is shown in green, and the neighborhood preserving robust region of $x$ is shown in pink. The former consists of points equidistant from $\mu^+, \mu^-$, and the latter consists of points equidistant from $x, \mu^+$.

## 3   The Neighborhood preserving Bayes optimal classifier

Robust classification is typically studied by setting the robustness regions, $\mathcal{U} = \{U_x\}_{x \in \mathbb{R}^d}$, to be balls of radius $r$ centered at $x$, $U_x = \{x' : \rho(x, x') \leq r\}$. The quantity $r$ is the robustness radius, and is typically set by the practitioner (before any training has occurred).

This method has a limitation with regards to trade-offs between accuracy and robustness. To increase the margin or robustness, we must have a large robustness radius (thus allowing us to defend from larger adversarial attacks). However, with large robustness radii, this can come at a cost of accuracy, as it is not possible to robustly give different labels to points with intersecting robustness regions.

For an illustration, consider Figure 1. Here we consider a data distribution $D = (\mu, \eta)$ in which the blue regions denote all points with $\eta(x) > 0.5$ (and thus should be labeled $+$), and the red regions denote all points with $\eta(x) < 0.5$ (and thus should be labeled $-$). Observe that it is not possible to be simultaneously accurate and robust at points $A, B$ while enforcing a large robustness radius, as demonstrated by the intersecting balls. While this can be resolved by using a smaller radius, this results in losing out on potential robustness at point $C$. In principal, we should be able to afford a large margin of robustness about $C$ due to its relatively far distance from the red regions.

Motivated by this issue, we seek to find a formalism for robustness that allows us to simultaneously avoid paying for any accuracy-robustness trade-offs and *adaptively* size robustness regions (thus allowing us to defend against a larger range of adversarial attacks at points that are located in more homogenous zones of the distribution support). To approach this, we will first provide an ideal limit object: a classifier that has the same accuracy as the Bayes optimal (thus meeting our first criteria) that has good robustness properties. We call this the the neighborhood preserving Bayes optimal classifier, defined as follows.

**Definition 2.** *Let $\mathcal{D} = (\mu, \eta)$ be a distribution over $\mathbb{R}^d \times \{\pm 1\}$. Then the **neighborhood preserving Bayes optimal classifier of** $\mathcal{D}$, denoted $g_{neighbor}$, is the classifier defined as follows. Let $\mu^+ = \{x : \eta(x) \geq \frac{1}{2}\}$ and $\mu^- = \{x : \eta(x) < \frac{1}{2}\}$. Then for any $x \in \mathbb{R}^d$, $g_{neighbor}(x) = +1$ if $\rho(x, \mu^+) \leq \rho(x, \mu^-)$, and $g_{neighbor}(x) = -1$ otherwise.*

This classifier can be thought of as the most robust classifier that matches the accuracy of the Bayes optimal. We call it *neighborhood preserving* because it extends the Bayes optimal classifier into a local neighborhood about every point in the support. For an illustration, refer to Figure 2, which plots the decision boundary of the neighborhood preserving Bayes optimal for an example distribution.

Next, we turn our attention towards measuring its robustness, which must be done with respect to some set of robustness regions $\mathcal{U} = \{U_x\}$. While these regions $U_x$ can be nearly arbitrary, we seek regions $U_x$ such that $A_{\mathcal{U}}(g_{max}, \mathcal{D}) = A(g_{bayes}, \mathcal{D})$ (our astuteness equals the maximum possible accuracy) and $U_x$ are "as large as possible" (representing large robustness). To this end, we propose the following regions.

**Definition 3.** *Let $\mathcal{D} = (\mu, \eta)$ be a data distribution over $\mathbb{R}^d \times \{\pm 1\}$. Let $\mu^+ = \{x : \eta(x) > \frac{1}{2}\}$, $\mu^- = \{x : \eta(x) < \frac{1}{2}\}$, and $\mu^{1/2} = \{x : \eta(x) = \frac{1}{2}\}$. For $x \in \mu^+$, we define the **neighborhood**

*preserving robustness region*, denoted $V_x$, as

$$V_x = \{x' : \rho(x, x') < \rho(\mu^- \cup \mu^{\frac{1}{2}}, x')\}.$$

*It consists of all points that are closer to $x$ than they are to $\mu^- \cup \mu^{1/2}$ (points oppositely labeled from $x$). We can use a similar definition for $x \in \mu^-$. Finally, if $x \in \mu^{1/2}$, we simply set $V_x = \{x\}$.*

These robustness regions take advantage of the structure of the neighborhood preserving Bayes optimal. They can essentially be thought of as regions that maximally extend from any point $x$ in the support of $\mathcal{D}$ to the decision boundary of the neighborhood preserving Bayes optimal. We include an illustration of the regions $V_x$ for an example distribution in Figure 2.

As a technical note, for $x \in supp(\mathcal{D})$ with $\eta(x) = 0.5$, we give them a trivial robustness region. The rational for doing this is that $\eta(x) = 0.5$ is an edge case that is arbitrary to classify, and consequently enforcing a robustness region at that point is arbitrary and difficult to enforce.

We now formalize the robustness and accuracy guarantees of the max-margin Bayes optimal classifier with the following two results.

**Theorem 4.** *(Accuracy) Let $\mathcal{D}$ be a data distribution. Let $\mathcal{V}$ denote the collection of neighborhood preserving robustness regions, and let $g$ denote the Bayes optimal classifier. Then the neighborhood preserving Bayes optimal classifier, $g_{neighbor}$, satisfies $A_{\mathcal{V}}(g_{neighbor}, \mathcal{D}) = A(g, \mathcal{D})$, where $A(g, \mathcal{D})$ denotes the accuracy of the Bayes optimal. Thus, $g_{neighbor}$ maximizes accuracy.*

**Theorem 5.** *(Robustness) Let $\mathcal{D}$ be a data distribution, let $f$ be a classifier, and let $\mathcal{U}$ be a set of robustness regions. Suppose that $A_{\mathcal{U}}(f, \mathcal{D}) = A(g, \mathcal{D})$, where $g$ denotes the Bayes optimal classifier. Then there exists $x \in supp(\mathcal{D})$ such that $V_x \not\subset U_x$, where $V_x$ denotes the neighborhood preserving robustness region about $x$. In particular, we cannot have $V_x$ be a strict subset of $U_x$ for all $x$.*

Theorem 4 shows that the neighborhood preserving Bayes classifier achieves maximal accuracy, while Theorem 5 shows that achieving a strictly higher robustness (while maintaining accuracy) is not possible; while it is possible to make accurate classifiers which have higher robustness than $g_{neighbor}$ in some regions of space, it is not possible for this to hold across all regions. Thus, the neighborhood preserving Bayes optimal classifier can be thought of as a local maximum to the constrained optimization problem of maximizing robustness subject to having maximum (equal to the Bayes optimal) accuracy.

## 3.1 Neighborhood Consistency

Having defined the neighborhood preserving Bayes optimal classifier, we now turn our attention towards building classifiers that converge towards it. Before doing this, we must precisely define what it means to converge. Intuitively, this consists of building classifiers whose robustness regions "approach" the robustness regions of the neighborhood preserving Bayes optimal classifier. This motivates the definition of *partial neighborhood preserving robustness regions*.

**Definition 6.** *Let $0 < \kappa < 1$ be a real number, and let $\mathcal{D} = (\mu, \eta)$ be a data distribution over $\mathbb{R}^d \times \{\pm 1\}$. Let $\mu^+ = \{x : \eta(x) > \frac{1}{2}\}$, $\mu^- = \{x : \eta(x) < \frac{1}{2}\}$, and $\mu^{1/2} = \{x : \eta(x) = \frac{1}{2}\}$. For $x \in \mu^+$, we define the **neighborhood preserving robustness region**, denoted $V_x$, as*

$$V_x = \{x' : \rho(x, x') < \kappa\rho(\mu^- \cup \mu^{\frac{1}{2}}, x')\}.$$

*It consists of all points that are closer to $x$ than they are to $\mu^- \cup \mu^{1/2}$ (points oppositely labeled from $x$) by a factor of $\kappa$. We can use a similar definition for $x \in \mu^-$. Finally, if $\eta(x) = \frac{1}{2}$, we simply set $V_x^\kappa = \{x\}$.*

Observe that $V_x^\kappa \subset V_x$ for all $0 < \kappa < 1$, and thus being robust with respect to $V_x^\kappa$ is a milder condition than $V_x$. Using this notion, we can now define margin consistency.

**Definition 7.** *A learning algorithm $A$ is said to be **neighborhood consistent** if the following holds for any data distribution $\mathcal{D}$. For any $0 < \epsilon, \delta, \kappa < 1$, there exists $N$ such that for all $n \geq N$, with probability at least $1 - \delta$ over $S \sim \mathcal{D}^n$,*

$$A_{\mathcal{V}^\kappa}(A_S, D) \geq A(g, \mathcal{D}) - \epsilon,$$

*where $g$ denotes the Bayes optimal classifier and $A_S$ denotes the classifier learned by algorithm $A$ from dataset $S$.*

This condition essentially says that the astuteness of the classifier learned by the algorithm converges towards the accuracy of the Bayes optimal classifier. Furthermore, we stipulate that this holds as long as the astuteness is measured with respect to some $\mathcal{V}^\kappa$. Observe that as $\kappa \to 1$, these regions converge towards the neighborhood preserving robustness regions, thus giving us a classifier with robustness effectively equal to that of the neighborhood preserving Bayes optimal classifier.

# 4  Neighborhood Consistent Non-Parametric Classifiers

Having defined neighborhood consistency, we turn to the following question: which non-parametric algorithms are neighborhood consistent? Our starting point will be the standard literature for the convergence of non-parametric classifiers with regard to accuracy. We begin by considering the standard conditions for $k_n$-nearest neighbors to converge (in accuracy) towards the Bayes optimal.

$k_n$-nearest neighbors is *consistent* if and only if the following two conditions are met: $\lim_{n\to\infty} k_n = \infty$, and $\lim_{n\to\infty} \frac{k_n}{n} = 0$. The first condition guarantees that each point is classified by using an increasing number of nearest neighbors (thus making the probability of a misclassification small), and the second condition guarantees that each point is classified using only points very close to it. We will refer to the first condition as *precision*, and the second condition as *locality*. A natural question is whether the same principles suffice for neighborhood consistency as well. We began by showing that without any additional constraints, the answer is no.

**Theorem 8.** *Let $\mathcal{D} = (\mu, \eta)$ be the data distribution where $\mu$ denotes the uniform distribution over $[0, 1]$ and $\eta$ is defined as: $\eta(x) = x$. Over this space, let $\rho$ be the euclidean distance metric. Suppose $k_n = O(\log n)$ for $1 \leq n < \infty$. Then $k_n$-nearest neighbors is not neighborhood consistent with respect to $\mathcal{D}$.*

The issue in the example above is that for smaller $k_n$, $k_n$-nearest neighbors lacks sufficient precision. For neighborhood consistency, points must be labeled using even more training points than are needed accuracy. This is because the classifier must be uniformly correct across the entirety of $V_x^\kappa$. Thus, to build neighborhood consistent classifiers, we must bolster the precision from the standard amount used for standard consistency. To do this, we begin by introducing *splitting numbers*, a useful tool for bolstering the precision of weight functions.

## 4.1  Splitting Numbers

We will now generalize beyond nearest neighbors to consider weight functions. Doing so will allow us to simultaneously analyze nearest neighbors and kernel classifiers. To do so, we must first rigorously substantiate our intuitions about increasing precision into concrete requirements. This will require several technical definitions.

**Definition 9.** *Let $\mu$ be a probability measure over $\mathbb{R}^d$. For any $x \in \mathbb{R}^d$, the **probability radius** $r_p(x)$ is the smallest radius for which $B(x, r_p(x))$ has probability mass at least $p$. More precisely, $r_p(x) = \inf\{r : \mu(B(x, r)) \geq p\}$.*

**Definition 10.** *Let $W$ be a weight function and let $S = \{x_1, x_2, \ldots, x_n\}$ be any finite subset of $\mathbb{R}^d$. For any $x \in \mathbb{R}^d$, $\alpha \geq 0$, and $0 \leq \beta \leq 1$, let $W_{x,\alpha,\beta} = \{i : \rho(x, x_i) \leq \alpha, w_i^S(x) \geq \beta\}$. Then the **splitting number** of $W$ with respect to $S$, denoted as $T(W, S)$ is the number of distinct subsets generated by $W_{x,\alpha\beta}$ as $x$ ranges over $\mathbb{R}^d$, $\alpha$ ranges over $[0, \infty)$, and $\beta$ ranges over $[0, 1]$. Thus $T(W, S) = |\{W_{x,\alpha,\beta} : x \in \mathbb{R}^d, 0 \leq \alpha, 0 \leq \beta \leq 1\}|$.*

Splitting numbers allow us to ensure high amounts of precision over a weight function. To prove neighborhood consistency, it is necessary for a classifier to be correct at *all* points in a given region. Consequently, techniques that consider a single point will be insufficient. The splitting number provides a mechanism for studying entire regions simultaneously. For more details on splitting numbers, we include several examples in the appendix.

## 4.2  Sufficient Conditions for Neighborhood Consistency

We now state our main result.

**Theorem 11.** *Let $W$ be a weight function, $\mathcal{D}$ a distribution over $\mathbb{R}^d \times \{\pm 1\}$, $\mathcal{U}$ a neighborhood preserving collection, and $(t_n)_1^\infty$ be a sequence of positive integers such that the following four conditions hold.*

*1. $W$ is consistent (with resp. to accuracy) with resp. to $\mathcal{D}$.*

*2. For any $0 < p < 1$, $\lim_{n \to \infty} E_{S \sim \mathcal{D}^n}[\sup_{x \in \mathbb{R}^d} \sum_1^n w_i^S(x) 1_{\rho(x, x_i) > r_p(x)}] = 0$.*

*3. $\lim_{n \to \infty} E_{S \sim D^n}[t_n \sup_{x \in \mathbb{R}^d} w_i^S(x)] = 0$.*

*4. $\lim_{n \to \infty} E_{S \sim D^n} \frac{\log T(W, S)}{t_n} = 0$.*

*Then $W$ is neighborhood consistent with respect to $\mathcal{D}$.*

**Remarks:** Condition 1 is necessary because neighborhood consistency implies standard consistency – or, convergence in accuracy to the Bayes Optimal. Standard consistency has been well studied for non-parametric classifiers, and there are a variety of results that can be used to ensure it – for example, Stone's Theorem (included in the appendix).

Conditions 2. and 3. are stronger version of conditions 2. and 3. of Stone's theorem. In particular, both include a supremum taken over all $x \in \mathbb{R}^d$ as opposed to simply considering a random point $x \sim \mathcal{D}$. This is necessary for ensuring correct labels on entire regions of points simultaneously. We also note that the dependence on $r_p(x)$ (as opposed to some fixed $r$) is a key property used for adaptive robustness. This allows the algorithm to adjust to potential differing distance scales over different regions in $\mathbb{R}^d$. This idea is reminiscent of the analysis given in [6], which also considers probability radii.

Condition 4. is an entirely new condition which allows us to simultaneously consider all $T(W, S)$ subsets of $S$. This is needed for analyzing weighted sums with arbitrary weights.

Next, we apply Theorem 11 to get specific examples of margin consistent non-parametric algorithms.

### 4.3 Nearest Neighbors and Kernel Classifiers

We now provide sufficient conditions for $k_n$-nearest neighbors to be neighborhood consistent.

**Corollary 12.** *Suppose $(k_n)_1^\infty$ satisfies (1) $\lim_{n \to \infty} \frac{k_n}{n} = 0$, and (2) $\lim_{n \to \infty} \frac{\log n}{k_n} = 0$. Then $k_n$-nearest neighbors is neighborhood consistent.*

As a result of Theorem 8, corollary 12 is tight for nearest neighbors. Thus $k_n$ nearest neighbors is neighborhood consistent if and only if $k_n = \omega(\log n)$.

Next, we give sufficient conditions for a kernel-similarity classifier.

**Corollary 13.** *Let $W$ be a kernel classifier over $\mathbb{R}^d \times \{\pm 1\}$ constructed from $K : \mathbb{R}^+ \to \mathbb{R}^+$ and $h_n$. Suppose the following properties hold.*

*1. $K$ is decreasing, and satisfies $\int_{\mathbb{R}^d} K(||x||) dx < \infty$.*

*2. $\lim_{n \to \infty} h_n = 0$ and $\lim_{n \to \infty} n h_n^d = \infty$.*

*3. For any $c > 1$, $\lim_{x \to \infty} \frac{K(cx)}{K(x)} = 0$.*

*4. For any $x \geq 0$, $\lim_{n \to \infty} \frac{n}{\log n} K(\frac{x}{h_n}) = \infty$.*

*Then $W$ is neighborhood consistent.*

Observe that conditions 1. 2. and 3. are satisfied by many common Kernel functions such as the Gaussian or Exponential kernel ($K(x) = \exp(-x^2)$/ $K(x) = \exp(-x)$). Condition 4. can be similarly satisfied by just increasing $h_n$ to be sufficiently large. Overall, this theorem states that Kernel classification is neighborhood consistent as long as the bandwidth shrinks slowly enough.

### 4.4 Histogram Classifiers

Having discussed neighborhood consistent nearest-neighbors and kernel classifier, we now turn our attention towards another popular weight function, histogram classifiers. Recall that histogram

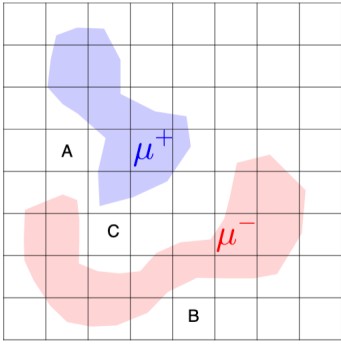

Figure 3: we have a histogram classifier being applied to the blue and red regions. The classifier will be unable to construct good labels in the cells labeled $A, B, C$, and consequently will not be robust with respect to $V_x^\kappa$ for sufficiently large $\kappa$.

classifiers operate by partitioning their input space into increasingly small cells, and then classifying each cell by using a majority vote from the training examples within that cell (a detailed description can be found in the appendix). We seek to answer the following question: is increasing precision sufficient for making histogram classifiers neighborhood consistent? Unfortunately, the answer this turns out not to be no. The main issue is that histogram classifiers have no mechanism for performing classification outside the support of the data distribution.

For an example of this, refer to Figure 3. Here we see a distribution being classified by a histogram classifier. Observe that the cell labeled $A$ contains points that are strictly closer to $\mu^+$ than $\mu^-$, and consequently, for sufficiently large $\kappa$, $V_x^\kappa$ will intersect $A$ for some point $x \in \mu^+$. A similar argument holds for the cells labeled $B$ and $C$.. However, since $A, B, C$ are all in cells that will never contain any data, they will never be labeled in a meaningful way. Because of this, histogram classifiers are not neighborhood consistent.

## 5    Validation

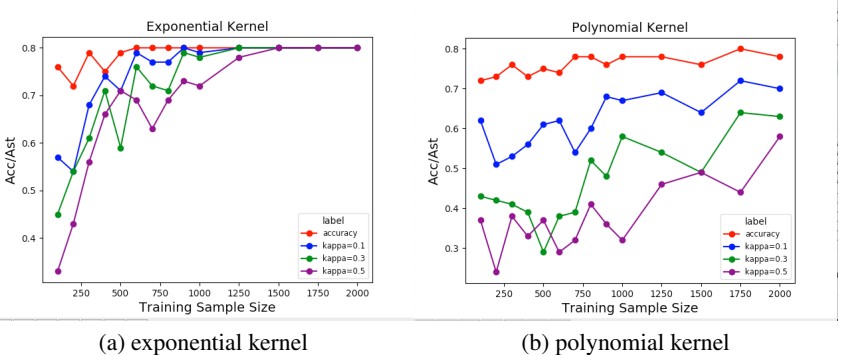

(a) exponential kernel          (b) polynomial kernel

Figure 4: Plots of astuteness against the training sample size. In both panels, accuracy is plotted in red, and the varying levels of robustness regions ($\kappa = 0.1, 0.3, 0.5$) are givne in blue, green and purple. In panel (a), observe that as sample size increases, every measure of astuteness converges towards $0.8$ which is as predicted by Corollary 13. In panel (b), although the accuracy appears to converge, none of the robustness measure. In fact, they get progressively worse the larger $\kappa$ gets.

To complement our theoretical large sample results for non-parametric classifiers, we now include several experiments to understand their behavior for finite samples. We seek to understand how quickly non-parametic classifiers converge towards the neighborhood preserving Bayes optimal.

We focus our attention on kernel classifiers and use two different kernel similarity functions: the first, an exponential kernel, and the second, a polynomial kernel. These classifiers were chosen so that the

former meets the conditions of Corollary 13, and the latter does not. Full details on these classifiers can be found in the appendix.

To be able to measure performance with increasing data size, we look at a simple synthetic dataset over overlayed circles (see Figure 5 for an illustration) with support designed so that the data is intrinsically multiscaled. In particular, this calls for different levels of robustness in different regions. For simplicity, we use a global label noise parameter of $0.2$, meaning that any sample drawn from this distribution is labeled differently than its support with probability $0.2$. Further details about our dataset are given in section D.

**Performance Measure.** For a given classifier, we evaluate its astuteness at a test point $x$ with respect to the robustness region $V_x^\kappa$ (Definition 6). While these regions are not computable in practice due to their dependency on the support of the data distribution, we are able to approximate them for this synthetic example due to our explicit knowledge of the data distribution. Details for doing this can be found in the appendix. To compute the empirical astuteness of a kernel classifier $W_K$ about test point $x$, we perform a grid search over all points in $V_x^\kappa$ to ensure that all points in the robustness region are labeled correctly.

For each classifier, we measure the empirical astuteness by using three trials of 20 test points and taking the average. While this is a relatively small amount of test data, it suffices as our purpose is to just verify that the algorithm roughly converges towards the optimal possible astuteness. Recall that for any neighborhood consistent algorithm, as $n \to \infty$, $A_{\mathcal{V}^\kappa}$ should converge towards $A^*$, the accuracy of the Bayes optimal classifier, for *any* $0 < \kappa < 1$. Thus, to verify this holds, we use $\kappa = 0.1, 0.3, 0.5$. For each of these values, we plot the empirical astuteness as the training sample size $n$ gets larger and larger. As a baseline, we also plot their standard accuracy on the test set.

**Results and Discussion:** The results are presented in Figure 4; the left panel is for the exponential kernel, while the right one is for the polynomial kernel. As predicted by our theory, we see that in all cases, the exponential kernel converges towards the maximum astuteness regardless of the value of $\kappa$: the only difference is that the rate of convergence is slower for larger values of $\kappa$. This is, of course, expected because larger values of $\kappa$ entail larger robustness regions.

By contrast, the polynomial kernel performs progressively worse for larger values of $\kappa$. This kernel was selected specifically to violate the conditions of Corollary 13, and in particular fails criteria 3. However, note that the polynomial kernel nevertheless performs will with respect to accuracy thus giving another example demonstrating the added difficulty of neighborhood consistency.

Our results bridge the gap between our asymptotic theoretical results and finite sample regimes. In particular, we see that kernel classifiers that meet the conditions of Corollary 13 are able to converge in astuteness towards the neighborhood preserving Bayes optimal classifier, while classifiers that do not meet these conditions fail.

## 6 Related Work

There is a wealth of literature on robust classification, most of which impose the same robustness radius $r$ on the entire data. [5, 17, 19, 20, 26, 15, 16, 18, 21, 22, 23], among others, focus primarily on neural networks, and robustness regions that are $\ell_1, \ell_2,$ or $\ell_\infty$ norm balls of a given radius $r$.

[7] and [12] show how to train neural networks with different robustness radii at different points by trading off robustness and accuracy; their work differ from ours in that they focus on neural networks, their robustness regions are still norm balls, and that their work is largely empirical.

Our framework is also related to large margin classification – in the sense that the robustness regions $\mathcal{U}$ induce a *margin constraint* on the decision boundary. The most popular large margin classifier is the Support Vector Machine[9, 3, 14] – a large margin linear classifier that minimizes the worst-case margin over the training data. Similar ideas have also been used to design classifiers that are more flexible than linear; for example, [27] shows how to build large margin Lipschitz classifiers by rounding globally Lipschitz functions. Finally, there has also been purely empirical work on achieving large margins for more complex classifiers – such as [13] for deep neural networks that minimizes the worst case margin, and [29] for metric learning to find large margin nearest neighbors. Our work differs from these in that our goal is to ensure a high enough local margin at each $x$, (by considering the neighborhood preserving regions $V_x$) as opposed to optimizing a global margin.

Finally, our analysis builds on prior work on robust classification for non-parametric methods in the standard framework. [1, 24, 28, 31] provide adversarial attacks on non-parametric methods. Wang et. al. [28] develops a defense for 1-NN that removes a subset of the training set to ensure higher robustness. Yang et. al [31] proposes the $r$-optimal classifier – which is the maximally astute classifier in the standard robustness framework – and proposes a defense called Adversarial Pruning.

Theoretically, [4] provide conditions under which weight functions converge towards the $r$-optimal classifier in the large sample limit. They show that for $r$-separated distributions, where points from different classes are at least distance $2r$ or more apart, nearest neighbors and kernel classifiers satisfy these conditions. In the more general case, they use Adversarial Pruning as a preprocessing step to ensure that the training data is $r$-separated, and show that this preprocessing step followed by nearest neighbors or kernel classifiers leads to solutions that are robust and accurate in the large sample limit. Our result fundamentally differs from theirs in that we analyze a different algorithm, and our proof techniques are quite different. In particular, the fundamental differences between the $r$-optimal classifier and the neighborhood preserving Bayes optimal classifier call for different algorithms and different analysis techniques.

In concurrent work, [8] proposes a similar limit to the neighborhood preserving Bayes optimal which they refer to as the margin canonical Bayes. However, their work then focuses on a data augmentation technique that leads to convergence whereas we focus on proving the neighborhood consistency of classical non-parametric classifiers.

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
