# A    Further Details of Definitions and Theorems

## A.1    Non-Parametric Classifiers

In this section, we precisely define weight functions, histogram classifiers and kernel classifiers.

**Definition 14.** *[11] A **weight function** $W$ is a non-parametric classifier with the following properties.*

1. *Given input $S = \{(x_1, y_1), (x_2, y_2,), \ldots, (x_n, y_n)\} \sim \mathcal{D}^n$, $W$ constructs functions $w_1^S, w_2^S, \ldots, w_n^S : \mathbb{R}^d \to [0, 1]$ such that for all $x \in \mathbb{R}^d$, $\sum_1^n w_i^S(x) = 1$. The functions $w_i^S$ are allowed to depend on $x_1, x_2, \ldots x_n$ but must be independent of $y_1, y_2, \ldots, y_n$.*

2. *$W$ has output $W_S$ defined as*

$$W_S(x) = \begin{cases} +1 & \sum_1^n w_i^S(x) y_i > 0 \\ -1 & \sum_1^n w_i^S(x) y_i \le 0 \end{cases}$$

   *As a result, $w_i^S(x)$ can be thought of as the weight that $(x_i, y_i)$ has in classifying $x$.*

**Definition 15.** *A **histogram classifier**, $H$, is a non-parametric classification algorithm over $\mathbb{R}^d \times \{\pm 1\}$ that works as follows. For a distribution $\mathcal{D}$ over $\mathbb{R} \times \{\pm 1\}$, $H$ takes $S = \{(x_i, y_i) : 1 \le i \le n\} \sim \mathcal{D}^n$ as input. Let $k_i$ be a sequence with $\lim_{i \to \infty} k_i = \infty$ and $\lim_{i \to \infty} \frac{k_i}{i} = 0$. $H$ constructs a set of hypercubes $C = \{c_1, c_2, \ldots, c_m\}$ as follows:*

1. *Initially $C = \{c\}$, where $S \subset c$.*

2. *For $c \in C$, if $c$ contains more than $k_n$ points of $S$, then partition $c$ into $2^d$ equally sized hypercubes, and insert them into $C$.*

3. *Repeat step 2 until all cubes in $C$ have at most $k_n$ points.*

*For $x \in \mathbb{R}$ let $c(x)$ denote the unique cell in $C$ containing $x$. If $c(x)$ doesn't exist, then $H_S(x) = -1$ by default. Otherwise,*

$$H_S(x) = \begin{cases} +1 & \sum_{x_i \in c(x)} y_i > 0 \\ -1 & \sum_{x_i \in c(x)} y_i \le 0 \end{cases}.$$

**Definition 16.** *A **partitioning rule** is a weight function $W$ over $\mathcal{X} \times \{\pm 1\}$ constructed in the following manner. Given $S = \{(x_i, y_i)\} \sim \mathcal{D}^n$, as a function of $\{x_1, \ldots, x_n\}$, we partition $\mathbb{R}^d$ into regions with $A(x)$ denoting the region containing $x$. Then, for any $x \in \mathbb{R}^d$ we have*

$$w_i^S(x) = \begin{cases} 1 & x_i \in A(x) \\ 0 & otherwise \end{cases}.$$

*To achieve $\sum w_i^S(x) = 1$, we can simply normalize weights for any $x$ by $\sum_1^n w_i^S(X)$.*

**Definition 17.** *A **kernel classifier** is a weight function $W$ over $\mathbb{R}^d \times \{\pm 1\}$ constructed from function $K : \mathbb{R}^+ \cup \{0\} \to \mathbb{R}^+$ and some sequence $\{h_n\} \subset \mathbb{R}^+$ in the following manner. Given $S = \{(x_i, y_i)\} \sim \mathcal{D}^n$, we have*

$$w_i^S(x) = \frac{K(\frac{\rho(x, x_i)}{h_n})}{\sum_{j=1}^n K(\frac{\rho(x, x_j)}{h_n})}.$$

*Then, as above, $W$ has output*

$$W_S(x) = \begin{cases} +1 & \sum_1^n w_i^S(x) y_i > 0 \\ -1 & \sum_1^n w_i^S(x) y_i \le 0 \end{cases}$$

## A.2    Splitting Numbers

We begin by restating definitions 9 and 10.

**Definition 9.** *Let $\mu$ be a probability measure over $\mathbb{R}^d$. For any $x \in \mathbb{R}^d$, the **probability radius** $r_p(x)$ is the smallest radius for which $B(x, r_p(x))$ has probability mass at least $p$. More precisely, $r_p(x) = \inf\{r : \mu(B(x, r)) \ge p\}$.*

**Definition 10.** *Let $W$ be a weight function and let $S = \{x_1, x_2, \ldots, x_n\}$ be any finite subset of $\mathbb{R}^d$. For any $x \in \mathbb{R}^d$, $\alpha \geq 0$, and $0 \leq \beta \leq 1$, let $W_{x,\alpha,\beta} = \{i : \rho(x, x_i) \leq \alpha, w_i^S(x) \geq \beta\}$. Then the **splitting number** of $W$ with respect to $S$, denoted as $T(W, S)$ is the number of distinct subsets generated by $W_{x,\alpha\beta}$ as $x$ ranges over $\mathbb{R}^d$, $\alpha$ ranges over $[0, \infty)$, and $\beta$ ranges over $[0, 1]$. Thus $T(W, S) = |\{W_{x,\alpha,\beta} : x \in \mathbb{R}^d, 0 \leq \alpha, 0 \leq \beta \leq 1\}|$.*

The main idea behind splitting numbers is that they allow us to ensure uniform convergence properties over a weight function. To prove neighborhood consistency, it is necessary for a classifier to be correct at *all* points in a given region. Consequently, techniques that consider a single point will be insufficient. The splitting number provides a mechanism for studying entire regions simultaneously. For clarity, we include a quick example in which we bound the splitting number for a given weight function.

**Example:** Let $W$ denote any kernel classifier corresponding such that $K : \mathbb{R}_{\geq 0} \to \mathbb{R}_{\geq 0}$ is a decreasing function. For any $S \sim \mathcal{D}^n$, observe that the condition $w_i^S(x) \geq \beta$ precisely corresponds to $\rho(x, x_i) \leq \gamma$ for some value of $\gamma$. This is because $w_i^S(x) > w_j^S(x)$ if and only if $\rho(x, x_i) < \rho(x, x_j)$. Thus, the regions $W_{x,\alpha,\beta}$ correspond to $\{i : \rho(x, x_i) \leq \gamma\}$, where $\gamma$ is a positive real number that depends on $x, \alpha, \beta$. These sets precisely correspond to subsets of $S$ that are contained within $B(x, \gamma)$. Since balls have VC dimension at most $d+2$, by Sauer's lemma, the number of subsets of $S$ that can be obtained in this manner is $O(n^{d+2})$. Therefore, we have that $T(W, S) = O(n^{d+2})$ for all $S \sim \mathcal{D}^n$.

### A.3 Stone's Theorem

**Theorem 18.** *[25] Let $W$ be weight function over $\mathbb{R}^d \times \{\pm 1\}$. Suppose the following conditions hold for any distribution $\mathcal{D}$ over $\mathbb{R}^d \times \{\pm 1\}$. Let $X$ be a random variable with distribution $\mathcal{D}_{\mathbb{R}^d}$, and $S = \{(x_1, y_1), (x_2, y_2), \ldots, (x_n, y_n)\} \sim \mathcal{D}^n$. All expectations are taken over $X$ and $S$.*

*1. There is a constant $c$ such that, for every nonnegative measurable function $f$ satisfying $\mathbb{E}[f(X)] < \infty$, and $\mathbb{E}[\sum_1^n w_i^S(X) f(x_i)] \leq c\mathbb{E}[f(x)]$.*

*2. $\forall a > 0$, $\lim_{n \to \infty} \mathbb{E}[\sum_1^n w_i^S(x) I_{||x_i - X|| > a||}] = 0$.*

*3. $\lim_{n \to \infty} \mathbb{E}[\max_{1 \leq i \leq n} w_i^S(X)] = 0$.*

*Then $W$ is consistent.*

## B Proofs

**Notation:**

- We let $\rho$ denote our distance metric over $\mathbb{R}^d$. For sets $X_1, X_2 \subset \mathbb{R}^d$, we let $\rho(X_1, X_2) = \inf_{x_1 \in X_1, x_2 \in X_2} \rho(x_1, x_2)$.

- For any $x \in \mathbb{R}^d$, $B(x, a) = \{x : \rho(x, x') \leq a\}$.

- For any measure over $\mathbb{R}^d$, $\mu$, we let $supp(\mu) = \{x : \mu(B(x, a)) > 0 \text{ for all } a > 0\}$.

- Given some measure $\mu$ over $\mathbb{R}^d$ and some $x \in \mathbb{R}^d$, we let $r_p(x)$ denote the probability radius (Definition 9) of $x$ with probability $p$. that is, $r_p(x) = \inf\{r : \mu(B(x, r)) \geq p\}$.

- For weight function $W$ and training sample $S$, we let $W_S$ denote the weight function learned by $W$ from $S$.

### B.1 Proofs of Theorems 4 and 5

*Proof.* (Theorem 4) Let $\mathcal{D} = (\mu, \eta)$ be a data distribution, and let $\mu^+, \mu^-$ be as described in section **??**. Observe that for any $x \in \mu^+$, the Bayes optimal classifier and the neighborhood preserving Bayes optimal both have the same output, and furthermore the neighborhood preserving Bayes gives this output (by definition) throughout the entirety of $V_x$, the neighborhood preserving robustness region of $x$. It follows that the neighborhood preserving Bayes optimal has optimal astuteness, as desired. $\qquad \square$

*Proof.* (Theorem 5) Let $\mathcal{D} = (\mu, \eta)$ be a data distribution, and assume towards a contradiction that there exists classifier $f$ which has maximal astuteness with respect towards some set of robustness regions $\mathcal{U} = \{U_x\}$ such that $V_x \subseteq U_x$ for all $x$. The key observation is that because $f$ has maximal astuteness, we must have $f(x) = g(x)$ for almost all points $x \sim \mu$ (where $g$ is the Bayes optimal classifier). Furthermore, for those values of $x$, we must have $g$ be robust at $x$ (meaning it uniformly outputs the same output through $U_x$).

In order for $U_x$ to be strictly larger than $V_x$ for some $x$, it *necessarily* must intersect with $U_{x'}$ for some $x'$ with $g(x') \neq g(x)$, and this is what causes the contradiction: $f$ cannot be astute at both $x$ and $x'$ if they are differently labeled and their robustness regions intersect. $\square$

## B.2    Proof of Theorem 8

Let $\mathcal{D} = (\mu, \eta)$ be the distribution with $\mu$ being the uniform distribution over $[0, 1]$ and $\eta : [0, 1] \to [0, 1]$ be $\eta(x) = x$. For example, if $(x, y) \sim \mathcal{D}$, then $\Pr[y = 1 | x = 0.3] = 0.3$.

We desire to show that $k_n$-nearest neighbors is not neighborhood consistent with respect to $\mathcal{D}$. We begin with the following key lemma.

**Lemma 19.** *For any $n > 0$, let $f_n$ denote the $k_n$-nearest neighbor classifier learned from $S \sim \mathcal{D}^n$. There exists some constant $\Delta > 0$ such that for all sufficiently large $n$, with probability at least $\frac{1}{2}$ over $S \sim \mathcal{D}^n$, there exists $x \in [0, 1]$ with $\frac{1}{2} - \Delta \leq x \leq \frac{1}{2} - \frac{3\Delta}{4}$ and $f_n(x) = +1$.*

*Proof.* Let $C$ be a constant such that $k_n \leq C \log n$ for all $2 \leq n < \infty$. Set $\Delta$ as

$$\frac{1}{2} \log_2 \frac{1}{1 - 2\Delta} + \frac{1}{2} \log_2 \frac{1}{1 + 2\Delta} < \frac{1}{C}. \tag{1}$$

Let $A \subset [0, 1]$ denote the interval $[\frac{1}{2} - \Delta, \frac{1}{2} - \frac{3\Delta}{4}]$. For $S \sim \mathcal{D}^n$, with high probability, there exist at least $\frac{\Delta n}{8}$ instances $x_i$ that are in $A$. Let us relabel these $x_i$ as $x_1, x_2, \ldots, x_m$ as

$$\frac{1}{2} - \Delta \leq x_1 < x_2 < \cdots < x_m \leq \frac{1}{2} - \frac{3\Delta}{4}.$$

Next, suppose that for some $i$, at least half of $y_i, y_{i+1}, \ldots, y_{i+k_n-1}$ are $+1$. Then it follows that $f_n(x) = +1$ for $x = \frac{x_{i+k_n} + x_i}{2}$ because the $k_n$ nearest neighbors of $x$ are precisely $x_i, x_{i+1}, \ldots x_{i+k_n-1}$ (as a technical note we make $x$ just slightly smaller to break the tie between $x_i$ and $x_{i+k_n}$). To lower bound the probability that this occurs for some $i$, we partition $y_1, y_2, \ldots y_m$ into at least $\frac{m}{2k_n}$ disjoint groups each containing $k_n$ consecutive values of $y_i$. We then bound the probability that each group will have at least $k_n/2$ +1s.

Consider any group of $k_n$ $y_i$s. We have that $\Pr[y_i] = +1 = \eta(x_i) = x_i \geq \frac{1}{2} - \Delta$. Since the variables $y_i$ are independent (even conditioning on $x_i$), it follows that the probability that at least half of them are +1 is at least $\Pr[\text{Bin}(k_n, \frac{1}{2} - \Delta) \geq \frac{k_n}{2}]$. For simplicity, assume that $k_n$ is even. Then using a standard lower bound for the tail of a binomial distribution (see, for example, Lemma 4.7.2 of [2]), we have that

$$\Pr[\text{Bin}(k_n, \frac{1}{2} - \Delta) \geq \frac{k_n}{2}] \geq \frac{1}{\sqrt{2k_n}} \exp(-k_n D(\frac{1}{2} || (\frac{1}{2} - \Delta))),$$

where $D(\frac{1}{2} || (\frac{1}{2} - \Delta)) = \frac{1}{2} \log_2 \frac{1}{1-2\Delta} + \frac{1}{2} \log_2 \frac{1}{1+2\Delta}$.

To simplify notation, let $D_\Delta = D(\frac{1}{2} || (\frac{1}{2} - \Delta))$. Then because we have $\frac{m}{2k_n}$ independent groups of $y_i$s, we have that

$$\Pr_{S \sim \mathcal{D}^n}[\exists x \in [\frac{1}{2} - \Delta, \frac{1}{2} - \frac{3\Delta}{4}] \text{ s.t. } f_n(x) = +1] \geq 1 - (1 - \frac{1}{\sqrt{2k_n}} \exp(-k_n D_\Delta))^{\frac{m}{2k_n}}$$

$$\geq 1 - \exp(-\frac{m}{2k_n \sqrt{2k_n}} e^{-k_n D_\Delta})$$

$$\geq 1 - \exp(-\frac{n\Delta}{(16C \log n)^{3/2}} e^{-C D_\Delta \log n}),$$

with the inequalities holding because $m \geq \frac{n\Delta}{8}$ and $k_n \leq C \log n$. By equation 1, $CD_\Delta < 1$. Therefore, $\lim_{n\to\infty} \frac{n}{(2C \log n)^{3/2}} e^{-CD_\Delta \log n} = \infty$, which implies that for $n$ sufficiently large,

$$\Pr_{S \sim \mathcal{D}^n}[\exists x \in [\frac{1}{2} - \Delta, \frac{1}{2} - \frac{3\Delta}{4}] \text{ s.t. } f_n(x) = +1] \geq \frac{1}{2},$$

as desired. $\square$

We now complete the proof of Theorem 8.

*Proof.* (Theorem 8) Let $\Delta$ be as described in Lemma 19, and let $\kappa = \frac{1}{2}$. For all $x < \frac{1}{2}$, we have that $[x, \frac{2x}{3} + \frac{1}{6}] \subseteq V_x^\kappa$. This is because we can easily verify that all points inside that interval are closer to $x$ than they are to $\frac{1}{2}$ (and consequently all points in $\mu^+ \cup \mu^{1/2}$) by factor of 2. It follows that for all $x \in [\frac{1}{2} - \frac{7\Delta}{8}, \frac{1}{2} - \Delta]$,
$$[\frac{1}{2} - \Delta, \frac{1}{2} - \frac{3\Delta}{4}] \subseteq V_x^\kappa.$$

However, applying Lemma 19, we know that with probability at least $\frac{1}{2}$, there exists some point $x' \in [\frac{1}{2} - \Delta, \frac{1}{2} - \frac{3\Delta}{4}]$ such that $f_n(x') = +1$. It follows that with probability at least $\frac{1}{2}$, $f_n$ lacks astuteness at *all* $x \in [\frac{1}{2} - \frac{7\Delta}{8}, \frac{1}{2} - \Delta]$. Since this set of points has total probability mass $\Delta/8$, it follows that with probability at least $\frac{1}{2}$, there is a fixed gap between $A_{\mathcal{V}^\kappa}(f_n, \mathcal{D})$ and $A(g, \mathcal{D})$ (as they differ in a region of probability mass at least $\Delta/8$). This implies that $k_n$-nearest neighbors is not neighborhood consistent. $\square$

## B.3 Proof of Theorem 11

Let $\mathcal{D} = (\mu, \eta)$ is a distribution over $\mathbb{R}^d \times \{\pm 1\}$. We will use the following notation: let $\mathcal{D}^+ = \{x : \eta(x) > \frac{1}{2}\}$, $\mathcal{D}^- = \{x : \eta(x) < \frac{1}{2}\}$ and $\mathcal{D}_{1/2} = \{x : \eta(x) = \frac{1}{2}\}$. In particular, we have that $\mathcal{D}^+ = \mu^+, \mathcal{D}^- = \mu^-$ and $\mathcal{D}_{1/2} = \mu^{1/2}$. This notation serve will be convenient throughout this section since it allows us to avoid overloading the symbol $\mu$.

To show that an algorithm is neighborhood consistent with respect to $\mathcal{D}$, we must show that for any $0 < \kappa < 1$, the astuteness with respect to $\mathcal{V}^\kappa$ converges towards the accuracy of the Bayes optimal. To this end, we fix any $0 < \kappa < 1$ and consider $\mathcal{V}^\kappa$.

For our proofs, it will be useful to have the additional assumption that the robustness regions, $V_x^\kappa$ are *closed*. To obtain this, we let $\mathcal{U} = \{U_x\}$ where $U_x = \overline{V_x^\kappa}$. Each $U_x$ is the closure of the corresponding $V_x^\kappa$, and in particular we have $V_x^\kappa \subset U_x$. Because of this, it will suffice for us to consider $A_\mathcal{U}$ as opposed to $A_{\mathcal{V}^\kappa}$ since $A_\mathcal{U}(f, \mathcal{D}) \leq A_{\mathcal{V}^\kappa}(f, \mathcal{D})$ for all classifiers $f$.

We now begin by first proving several useful properties of $\mathcal{U}$ that we will use throughout this entire section.

**Lemma 20.** *The collection of sets $\mathcal{U} = \{U_x\}$ defined as $U_x = \overline{V_x^\kappa}$ satisfies the following properties.*

1. *$U_x$ is closed for all $x$.*

2. *if $x \in \mathcal{D}^+$, for all $x' \in U_x$, $\rho(x, x') < \rho(\mathcal{D}^+ \cup \mathcal{D}_{1/2}, x')$.*

3. *if $x \in \mathcal{D}^-$, for all $x' \in U_x$, $\rho(x, x') < \rho(\mathcal{D}^- \cup \mathcal{D}_{1/2}, x')$.*

4. *$U_x = \{x\}$ for all $x \in \mathcal{D}_{1/2}$.*

5. *$U_x$ is bounded for all $x$.*

*Here $\mu^+, \mu^-, \mu^{1/2}$ are as described in section ??.*

*Proof.* Property (1) is given the by definition, and properties (2), (3) follow from the fact that $\kappa$ is strictly less than 1. In particular, the distance function $\rho$ is continuous and consequently all limit points of a set have distances that are limits of distances within the set. Property (4) is since $V_x^\kappa = \{x\}$ for all $x \in \mathcal{D}_{1/2}$.

Finally, property (5) follows from the fact that $\kappa < 1$. As $x$ gets arbitrarily far away from $x$ the ratio of its distance to $x$ with its distance to $\mu^-$ gets arbitrarily close to 1, and consequently there is some maximum radius $R$ so that $V_x^\kappa \subset B(x, R)$. Since $B(x, R)$ is closed, it follows that $U_x \subset B(x, R)$ as well. $\qquad\square$

Next, fix $W$ as a weight function and $t_n$ is a sequence of positive integers such that the conditions of Theorem 11 hold, that is:

1. $W$ is consistent (with resp. to accuracy) with resp. to $\mathcal{D}$.

2. For any $0 < p < 1$, $\lim_{n\to\infty} E_{S\sim\mathcal{D}^n}[\sup_{x\in\mathbb{R}^d} \sum_1^n w_i^S(x)1_{\rho(x,x_i)>r_p(x)}] = 0$.

3. $\lim_{n\to\infty} E_{S\sim D^n}[t_n \sup_{x\in\mathbb{R}^d} w_i^S(x)] = 0$.

4. $\lim_{n\to\infty} E_{S\sim D^n} \frac{\log T(W,S)}{t_n} = 0$.

Finally, we will also make the additional assumption that $\mathcal{D}$ has infinite support. Cases where $\mathcal{D}$ has finite support can be somewhat trivially handled: when the sample size goes to infinity, we will have perfect labels for every point in the support, and consequently condition 2. will ensure that any $x' \in V_x^\kappa$ is labeled according to the label of $x$.

We also use the following notation. For any classifier $f : \mathbb{R}^d \to \{\pm 1\}$, we let

$$\mathcal{D}_f^+ = \{x : f(x' = +1 \text{ for all } x' \in U_x\}, \text{ and } \mathcal{D}_f^- = \{x : f(x' = -1 \text{ for all } x' \in U_x\}. \quad (2)$$

These sets represent the examples that $f$ robustly labels as $+1$ and $-1$ respectively. These sets are useful since they allows us to characterize the astuteness of $f$, which we do with the following lemma.

**Lemma 21.** *For any classifier $f : \mathbb{R}^d \to \{\pm 1\}$, we have*

$$A_{\mathcal{U}}(f, \mathcal{D}) \geq A(g, \mathcal{D}) - \mu(\mathcal{D}^+ \setminus \mathcal{D}_f^+) - \mu(D^- \setminus \mathcal{D}_f^-),$$

*where $g$ denotes the Bayes optimal classifier.*

*Proof.* By property 4 of Lemma 20, $U_x = \{x\}$ for all $x \in \mathcal{D}_{1/2}$. Consequently, if $x \in \mathcal{D}_{1/2}$, there is a $\frac{1}{2}$ chance that any classifier is astute at $(x, y)$. Using this along with the definition of astuteness, we see that

$$A_{\mathcal{U}}(f, \mathcal{D}) = \Pr_{(x,y)\sim\mathcal{D}}[f(x') = y \text{ for all } x' \in U_x]$$

$$= \Pr_{(x,y)\sim\mathcal{D}}[y = +1 \text{ and } x \in (D^+ \cap D_f^+)] + \Pr_{(x,y)\sim\mathcal{D}}[y = -1 \text{ and } x \in (D^- \cap D_f^-)] + \frac{1}{2}\Pr_{(x,y)\sim\mathcal{D}}[x \in \mathcal{D}_{1/2}]$$

However, observe by the definitions of $\mathcal{D}^+, \mathcal{D}^-$ and $\mathcal{D}_{1/2}$ that

$$A(g, \mathcal{D}) = \Pr_{(x,y)\sim\mathcal{D}}[y = +1 \text{ and } x \in D^+] + \Pr_{(x,y)\sim\mathcal{D}}[y = -1 \text{ and } x \in D^-] + \frac{1}{2}\Pr_{(x,y)\sim\mathcal{D}}[x \in \mathcal{D}_{1/2}].$$

Substituting this, we find that

$$A_{\mathcal{U}}(f, \mathcal{D}) \geq A(g, \mathcal{D}) - \Pr_{(x,y)\sim\mathcal{D}}[x \in (D^+ \setminus D_f^+)] - \Pr_{(x,y)\sim\mathcal{D}}[x \in (D^- \setminus D_f^-)]$$

$$= A(g, \mathcal{D}) - \mu(\mathcal{D}^+ \setminus \mathcal{D}_f^+) - \mu(D^- \setminus \mathcal{D}_f^-),$$

as desired. $\qquad\square$

Lemma 21 shows that to understand how $W_S$ converges in astuteness, it suffices to understand how the regions $\mathcal{D}_{W_S}^+$ and $\mathcal{D}_{W_S}^-$ converge towards $D^+$ and $D^-$ respectively. This will be our main approach for proving Theorem 11. Due to the inherent symmetry between $+$ and $-$, we will focus on showing how the region $\mathcal{D}_{W_S}^+$ converges towards $D^+$. The case for $-$ will be analogous. To that end, we have the following key definition.

**Definition 22.** *Let $p, \Delta > 0$. We say $x \in \mathcal{D}^+$ is $(p, \Delta)$-**covered** if for all $x' \in U_x$ and for all $x'' \in B(x', r_p(x')) \cap supp(\mu)$, $\eta(x'') > \frac{1}{2} + \Delta$. Here $r_p$ denotes the probability radius (Definition 9). We also let $\mathcal{D}_{p,\Delta}^+$ denote the set of all $x \in \mathcal{D}^+$ that are $(p, \Delta)$-covered.*

If $x$ is $(p, \Delta)$-covered, it means that for all $x' \in U_x$, there is a set of points with measure $p$ around $x'$ that are both close to $x'$, and likely (with at least probability $\frac{1}{2} + \Delta$) to be labeled as $+1$. Our main idea will be to show that if $x$ is $(p, \Delta)$ covered and $n$ is sufficiently large, $x$ is likely to be in $\mathcal{D}_{W_S}^+$.

We begin this process by first showing that all $x$ are $(p, \Delta)$-covered for some $p, \Delta$. To do so, it will be useful to have one more piece of notation which we will also use throughout the rest of the section. We let

$$\mathcal{D}_{1/2}^- = \mathcal{D}^- \cup \mathcal{D}_{1/2} = supp(\mu) \setminus \mathcal{D}^+.$$

This set will be useful, since Lemma 20 implies that for all $x \in \mathcal{D}^+$ and for all $x' \in U_x$, $\rho(x, x') < \rho(\mathcal{D}_{1/2}^-, x')$. We now return to showing that all $x$ are $(p, \Delta$-covered for some $p, \Delta$.

**Lemma 23.** *For any $x \in \mathcal{D}^+$, there exists $p, \Delta > 0$ such that $x$ is $(p, \Delta)$-covered.*

*Proof.* Fix any $x$. Let $f : U_x \to \mathbb{R}$ be the function defined as $f(x') = \rho(x', \mathcal{D}_{1/2}^-) - \rho(x', x)$. Observe that $f$ is continuous. By assumption, $U_x$ is closed and bounded, and consequently must attain its minimum. However, by Lemma 20, we have that $f(x') > 0$ for all $x' \in U_x$. it follows that $\min_{x' \in U_x} f(x') = \gamma$ where $\gamma > 0$.

Next, let $p = \mu(B(x, \gamma/2))$. $p > 0$ since $x \in supp(\mu)$. Observe that for any $x' \in U_x$, $r_p(x') \leq \rho(x, x') + \gamma/2$, where, $r_p(x')$ denotes the probability radius of $x'$. This is because $B(x', (\rho(x, x') + \gamma/2))$ contains $B(x, \gamma/2)$ which has probability mass $p$. It follows that for any $x' \in U_x$, $\rho(x', \mathcal{D}_{1/2}^-) \geq r_p(x') + \gamma/2$. Motivated by this observation, let $A$ be the region defined as

$$A = \bigcup_{x' \in U_x} B(x', r_p(x')).$$

Then by our earlier observation, we have that $\rho(A, \mathcal{D}_{1/2}^-) \geq \frac{\gamma}{2}$. Since distance is continuous, it follows that $\rho(\overline{A}, \mathcal{D}_{1/2}^-) \geq \frac{\gamma}{2}$ as well, where $\overline{A}$ denotes the closure of $A$.

This means that for any $x'' \in \overline{A} \cap supp(\mu)$, $\eta(x'') > \frac{1}{2}$, since otherwise $\rho(\overline{A}, \mathcal{D}_{1/2}^-)$ would equal $0$ (as the two sets would literally intersect). Finally, $supp(\mu)$ is a closed set (see Appendix C.1), and thus $\overline{A} \cap supp(\mu)$ is closed as well. Since $\eta$ is continuous (by assumption from Definition **??**), it follows that $\eta$ must maintain its minimum value over $\overline{A} \cap supp(\mu)$. It follows that there exists $2\Delta > 0$ such that $\eta(x'') \geq \frac{1}{2} + 2\Delta > \frac{1}{2} + \Delta$ for all $x'' \in \overline{A} \cap supp(\mu)$.

Finally, by the definition of $A$, for all $x' \in U_x$, $B(x', r_p(x')) \subset A$. It consequently follows from the definition that $x$ is $(p, \Delta)$-covered, as desired. $\square$

While the previous lemma show that some $p, \Delta$ cover any $x \in \mathcal{D}^+$, this does not necessarily mean that there are some fixed $p, \Delta$ that cover *all* $x \in \mathcal{D}^+$. Nevertheless, we can show that this is almost true, meaning that there are some $p, \Delta$ that cover *most* $x \in \mathcal{D}^+$. Formally, we have the following lemma.

**Lemma 24.** *For any $\epsilon > 0$, there exists $p, \Delta$ such that $\mu(\mathcal{D}^+ \setminus \mathcal{D}_{p,\Delta}^+) < \epsilon$, where $\mathcal{D}_{p,\Delta}^+$ is as defined in Definition 22.*

*Proof.* Observe that if $x$ is $(p, \Delta)$-covered, then it is also $(p', \Delta')$-covered for any $p' < p$ and $\Delta' < \Delta$. This is because $B(x', r_{p'}(x')) \subset B(x', r_p(x))$ and $\frac{1}{2} + \Delta > \frac{1}{2} + \Delta'$. Keeping this in mind, define

$$\mathcal{A} = \{\mathcal{D}_{1/i, 1/j}^+ : i, j \in \mathbb{N}\}.$$

For any $x \in \mathcal{D}^+$, by Lemma 23 and our earlier observation, there exists $A \in \mathcal{A}$ such that $x \in A$. It follows that $\cup_{A \in \mathcal{A}} A = \mathcal{D}^+$. By applying Lemma 41, we see that there exists a finite subset of $\mathcal{A}$, $\{A_1, \ldots, A_m\}$ such that

$$\mu(A_1 \cup \cdots \cup A_m\}) > \mu(\mathcal{D}^+) - \epsilon.$$

Let $A_k = \mathcal{D}_{1/i_k, 1/j_k}^+$ for $1 \leq k \leq m$. From our previous observation once again, we see that $\cup A_i \subset \mathcal{D}_{1/I, 1/J}^+$ where $I = \max(i_k)$ and $J = \max(j_k)$. It follows that setting $p = 1/I$ and $\Delta = 1/J$ suffices. $\square$

Recall that our overall goal is to show that if $x$ is $(p, \Delta)$-covered, $n$ is sufficiently large, then $x$ is very likely to be in $\mathcal{D}^+_{W_S}$ (defined in equation 2). To do this, we will need to find sufficient conditions on $S$ for $x$ to be in $W_S$. This requires the following definitions, that are related to *splitting numbers* (Definition 10).

**Definition 25.** *Let $x \in \mathbb{R}^d$ be a point, and let $S = \{(x_1, y_1), \ldots, (x_n, y_n)\}$ be a training set sampled from $\mathcal{D}^n$. For $0 \le \alpha$, $0 \le \beta \le 1$, and $0 < \Delta < \frac{1}{2}$, we define*

$$W^{\Delta, S}_{x, \alpha, \beta} = \{i : \rho(x, x_i) \le \alpha, w^S_i(x) \ge \beta, \eta(x_i) > \frac{1}{2} + \Delta\}.$$

**Definition 26.** *Let $0 < \Delta < \frac{1}{2}$, and let $S = \{(x_1, y_1), \ldots, (x_n, y_n)\}$ be a training set sampled from $\mathcal{D}^n$. Then we let*

$$W^{\Delta, S} = \{W^{\Delta, S}_{x, \alpha, \beta} : x \in \mathbb{R}^d, 0 \le \alpha, 0 \le \beta \le 1\}.$$

These convoluted looking sets will be useful for determining the behavior of $W_s$ at some $x \in \mathcal{D}^+_{p, \Delta}$. Broadly speaking, the idea is that if every set of indices $R \subset W^{\Delta, S}$ is relatively well behaved (i.e. the number of $y_i$s that are $+1$ is close to $(|R|(\frac{1}{2} + \Delta)$, the expected amount), then $W_s(x') = +1$ for all $x' \in U_x$. Before showing this, we will need a few more lemmas.

**Lemma 27.** *Fix any $\delta > 0$ and let $0 < \Delta < \frac{1}{2}$. There exists $N$ such that for all $n > N$ the following holds. With probability $1 - \delta$ over $S \sim \mathcal{D}^n$, for all $R \in W^{\Delta, S}$ with $|R| > t_n$, $\frac{1}{|R|} \sum_{i \in R} y_i \ge \Delta$*

*Proof.* The key idea is to observe that the set $W^{\Delta, S}$ and the value $T(W, S)$ are completely determined by $\{x_1, \ldots, x_n\}$. This is because weight functions choose their weights only through dependence on $x_1, \ldots, x_n$. Consequently, we can take the equivalent formulation of first drawing $x_1, \ldots, x_n \sim \mu^n$, and then drawing $y_i$ independently according to $y_i = 1$ with probability $\eta(x_1)$ and 0 with probability $1 - \eta(x_i)$. In particular, we can treat $y_1, \ldots, y_n$ as independent from $W^{\Delta, S}$ and $T(W, S)$ conditioning on $x_1, \ldots, x_n$.

Fix any $x_1, \ldots, x_n$. First, we see that $|W^{\Delta, S}| \le T(W, S)$. This is because $W^{\Delta, S}_{x, \alpha, \beta}$ is a subset that is uniquely defined by $W_{x, \alpha, \beta}$ (see Definitions 25 and 10). Second, for any $R \in W^{\Delta, S}$, observe that for all $i \in R$, $y_i$ is a binary variable in $[-1, 1]$ with expected value at least $(\frac{1}{2} + \Delta) - (\frac{1}{2} - \Delta) = 2\Delta$ (again by the definition). It follows that if $|R| \ge t_n$, by Hoeffding's inequality

$$\Pr_{y_1 \ldots y_n} [\sum_{i \in R} y_i < \Delta] \le \exp\left(-\frac{2|R|^2 \Delta^2}{4|R|}\right) \le \exp\left(-\frac{t_n \Delta^2}{2}\right).$$

Since there at most $T(W, S)$ sets $R$, it follows that

$$\Pr_{y_1 \ldots y_n} [\sum_{i \in R} y_i < \Delta \text{ for some } R \in W^{\Delta, S} \text{ with } |R| > t_n] \le T(W, S) \exp\left(-\frac{t_n \Delta^2}{2}\right).$$

However, by condition 4. of Theorem 11, it is not difficult to see that this quantity has expectation that tends to 0 as $n \to \infty$ (unless $T(W, S)$ uniformly equals 1, but this degenerate case can easily be handled on its own). Thus, for any $\delta > 0$, it follows that there exists $N$ such that for all $n > N$, with probability at least $1 - \frac{\delta}{2}$, $T(W, S) \exp\left(-\frac{t_n \Delta^2}{2}\right) \le \frac{\delta}{2}$. This value of $N$ consequently suffices for our lemma. $\square$

We now relate $\mathcal{D}^+_{W_S}$ (Equation 2) to $W^{\Delta, S}$ as well as the conditions of Theorem 11.

**Lemma 28.** *Let $S = \{(x_1, y_1), \ldots, (x_n, y_n)\}$ and let $0 < \Delta \le \frac{1}{2}$ and $0 < p < 1$ such that the following conditions hold.*

1. *For all $R \in W^{\Delta, S}$ with $|R| > t_n$, $\frac{1}{|R|} \sum_{i \in R} y_i \ge \Delta$.*

2. *$\sup_{x \in \mathbb{R}^d} \sum_1^n w^S_i(x) 1_{\rho(x, x_i) > r_p(x)} < \frac{\Delta}{5}$.*

3. *$t_n \sup_{x \in \mathbb{R}^d} w^S_i(x) < \frac{\Delta}{5}$.*

*Then $\mathcal{D}^+_{p, \Delta} \subseteq \mathcal{D}^+_{W_S}$.*

*Proof.* Let $x \in \mathcal{D}_{p,\Delta}^+$, and let $x' \in U_x$ be arbitrary. It suffices to show that $W_S(x') = +1$ (as $x, x'$ were arbitrarily chosen). From the definition of $W_S$, this is equivalent to showing that $\sum_1^n w_i^S(x')y_i > 0$. Thus, our strategy will be to lower bound this sum using the conditions given in the lemma statement.

We first begin by simplifying notation. Since $S$ and $x'$ are both fixed, we use $w_i$ to denote $w_i^S(x')$. Since $n$ is fixed, we will also use $t$ to denote $t_n$. Next, suppose that $|\{x_1, \ldots, x_n\} \cap B(x', r_p(x'))| = k$. Without loss of generality, we can rename indices such that $\{x_1, \ldots, x_n\} \cap B(x', r_p(x')) \cap B(x', r_p(x')) = \{x_1, \ldots, x_k\}$, and $w_1 \geq w_2 \geq \cdots \geq w_k$.

Let $Y_j = \sum_{i=1}^j y_i$. Our main idea will be to express the sum in terms of these $Y_j$s as follows.

$$\sum_1^n w_i y_i = \sum_1^k w_i y_i + \sum_{k+1}^n w_i y_i$$

$$= w_k Y_k + (w_{k-1} - w_k)Y_{k-1} + \cdots + (w_{t+1} - w_{t+2})Y_{t+1} + \sum_{i=1}^t (w_i - w_{t+1})y_i + \sum_{k+1}^n w_i y_i$$

$$= \underbrace{w_k Y_k + \sum_{i=t+1}^{k-1}(w_i - w_{i+1})Y_i}_{\alpha} + \underbrace{\sum_{i=1}^t (w_i - w_{t+1})y_i}_{\beta} + \underbrace{\sum_{k+1}^n w_i y_i}_{\tau}.$$

We now bound $\alpha, \beta$ and $\tau$ in terms of $\Delta$ by using the conditions given in the lemma. We begin with $\beta$ and $\tau$, which are considerably easier to handle.

For $\beta$, we have that

$$\beta = \sum_{i=1}^t (w_i - w_{t+1})y_i \geq \sum_{i=1}^t (w_i - w_{t+1})(-1) \geq -tw_1.$$

By condition 2 of the lemma, we see that $tw_1 < \frac{\Delta}{5}$, which implies that $\beta \geq -\frac{\Delta}{5}$.

For $\gamma$, we have that $\gamma = \sum_{k+1}^n w_i y_i \geq -\sum_{k+1}^n w_i$. However, for all $k+1 \leq i \leq n$, by definition of $k$, $\rho(x', x_i) > r_p(x')$. It follows from condition 3 of the lemma that $\gamma \geq -\frac{\Delta}{5}$.

Finally, we handle $\alpha$. Recall that $x$ is $(p, \Delta)$-covered. It follows that for all $x'' \in supp(\mu) \cap B(x', r_p(x'))$, $\eta(x'') > \frac{1}{2} + \Delta$. Thus, by the definition of $k$, $\eta(x_i) > \frac{1}{2} + \Delta$ for $1 \leq i \leq k$. It follows that if $w_i > w_{i+1}$ or $i = k$, then

$$W_{x', r_p(x'), w_i}^{\Delta, S} = \{j : \rho(x', x_j) \leq r_p(x'), w_j \geq w_i, \eta(x_j) > \frac{1}{2} + \Delta\}$$

$$= \{1, \ldots, i\}.$$

This implies that $\{1, \ldots, i\} \in W^{\Delta, S}$, and consequently that $Y_i \geq i\Delta$, from condition 1 of the lemma. It follows that for all $t < i \leq k$, $(w_i - w_{i+1})Y_i \geq i(w_i - w_{i+1})\Delta$, and that $w_k Y_k \geq kw_k \Delta$. Substituting these, we find that

$$\alpha = w_k Y_k + \sum_{i=t+1}^{k-1}(w_i - w_{i+1})Y_i$$

$$\geq kw_k \Delta + \sum_{i=t+1}^{k-1} i(w_i - w_{i+1})\Delta$$

$$= w_k \Delta + w_{k-1}\Delta + \cdots + w_{t+1}\Delta + (t+1)w_{t+1}\Delta.$$

$$\geq (1 - \sum_{1^t} w_i - \sum_{k+1}^n w_i)\Delta$$

$$\geq (1 - \frac{2\Delta}{5})\Delta$$

$$\geq (\frac{4\Delta}{5}),$$

with the last inequalities holding from the arguments given for $\beta$ and $\gamma$ along with the fact that $0 < \Delta \leq \frac{1}{2}$. Finally, substituting these, we find that $\alpha + \beta + \gamma \geq \frac{4\Delta}{5} - \frac{2\Delta}{5} = \frac{2\Delta}{5} > 0$, as desired. $\qquad \square$

We are now ready to prove the key lemma that forms one half of the main theorem (the other half corresponding to $\mathcal{D}_{W_S}^-$).

**Lemma 29.** *Let $\delta, \epsilon > 0$. There exists $N$ such that for all $n > N$, with probability $1 - \delta$ over $S \sim \mathcal{D}^n$, $\mu(\mathcal{D}^+ \setminus \mathcal{D}_{W_S}^+) < \epsilon$.*

*Proof.* First, by Lemma 24, let $0 < p$ and $0 < \Delta$ be such that $\mu(\mathcal{D}^+ \setminus \mathcal{D}_{p,\Delta}^+) < \epsilon$. By combining Lemma 27, condition 3 of Theorem 11, and condition 2 of Theorem 11 respectively, we see that there exists $N$ such that for all $n > N$, the following hold:

1. With probability at least $1 - \frac{\delta}{3}$ over $S \sim \mathcal{D}^n$, for all $R \in W^{\Delta,S}$ with $|R| > t_n$, $\frac{1}{|R|} \sum_{i \in R} y_i \geq \Delta$.

2. With probability at least $1 - \frac{\delta}{3}$ over $S \sim \mathcal{D}^n$, $\sup_{x \in \mathbb{R}^d} \sum_1^n w_i^S(x) 1_{\rho(x,x_i) > r_p(x)} < \frac{\Delta}{5}$.

3. With probability at least $1 - \frac{\delta}{3}$ over $S \sim \mathcal{D}^n$, $t_n \sup_{x \in \mathbb{R}^d} w_i^S(x) < \frac{\Delta}{5}$.

By a union bound, this implies that $p, \Delta, S$ satisfy the conditions of Lemma 28 with probability at least $1 - \delta$. Thus, applying the Lemma, we see that with probability $1 - \delta$, $\mathcal{D}_{p,\Delta}^+ \subset \mathcal{D}_{W_S}^+$. This immediately implies our claim. $\qquad \square$

By replicating all of the work in this section for $\mathcal{D}^-$ and $\mathcal{D}_{p,\Delta}^-$, we can similarly show the following:

**Lemma 30.** *Let $\delta, \epsilon > 0$. There exists $N$ such that for all $n > N$, with probability $1 - \delta$ over $S \sim \mathcal{D}^n$, $\mu(\mathcal{D}^- \setminus \mathcal{D}_{W_S}^-) < \epsilon$.*

Combining these two lemmas with Lemma 21 immediately implies that for all $\delta, \epsilon > 0$, there exists $N$ such that for all $n > N$, with probability $1 - \delta$ over $S \sim \mathcal{D}^n$,

$$A_{\mathcal{U}}(W_S, \mathcal{D}) \geq A(g, \mathcal{D}) - \epsilon.$$

Since $V_x^\kappa \subset U_x$ and since $\kappa$ was arbitrary, this implies Theorem 11, which completes our proof.

### B.4 Proof of Corollary 12

Recall that $k_n$-nearest neighbors can be interpreted as a weight function, in which $w_i^S(x) = \frac{1}{k_n}$ if $x_i$ is one of the $k_n$ closest points to $x$, and 0 otherwise. Therefore, it suffices to show that the conditions of Theorem 11 are met.

We let $W$ denote the weight function associated with $k_n$-nearest neighbors.

**Lemma 31.** *$W$ is consistent.*

*Proof.* It is well known (for example [6]) that $k_n$-nearest neighbors is consistent for $\lim_{n \to \infty} k_n = \infty$ and $\lim_{n \to \infty} \frac{k_n}{n} = 0$. These can easily be verified for our case. $\qquad \square$

**Lemma 32.** *For any $0 < p < 1$, $\lim_{n \to \infty} \mathbb{E}_{S \sim \mathcal{D}^n}[\sup_{x \in \mathbb{R}^d} \sum_1^n w_i^S(x) 1_{\rho(x,x_i) > r_p(x)}] = 0$.*

*Proof.* It suffices to show that for $n$ sufficiently large, all $k_n$-nearest neighbors of $x$ are located inside $B(x, r_p(x))$ for all $x \in \mathbb{R}^d$. We do this by using a VC-dimension type argument to show that all balls $B(x, r)$ contain a number of points from $S \sim \mathcal{D}^n$ that is close to their expectation.

For $x \in \mathbb{R}^d$ and $r \geq 0$, let $f_{x,r}$ denote the $0 - 1$ function defined as $f_{x,r}(x') = 1_{x' \in B(x,r)}$. Let $F = \{f_{x,r} : x \in \mathbb{R}^d, r \geq 0\}$ denote the class of all such functions. It is well known that the VC dimension of $F$ is at most $d + 2$.

For $f \in F$, let $\mathbb{E}f$ denote $\mathbb{E}_{(x',y) \sim \mathcal{D}} f(x')$ and $\mathbb{E}_n f$ denote $\frac{1}{n} \sum_1^n f(x_i)$, where $\mathbb{E}_n f$ is defined with respect to some sample $S \sim \mathcal{D}^n$. By the standard generalization result of Vapnik and Chervonenkis (see [10] for a proof), we have that with probability $1 - \delta$ over $S \sim \mathcal{D}^n$,

$$- \beta_n \sqrt{\mathbb{E}f} \leq \mathbb{E}f - \mathbb{E}_n f \leq \beta_n \sqrt{\mathbb{E}f} \tag{3}$$

holds for all $f \in F$, where $\beta_n = \sqrt{(4/n)((d+2)\ln 2n + \ln(8/\delta))}$.

Suppose $n$ is sufficiently large so that $\beta_n \leq \frac{p}{2}$ and $\frac{k_n}{n} < \frac{p}{2}$, and suppose that equation 3 holds. Pick any $x \in \mathbb{R}^d$ and consider $f_{x,r}$ where $r > r_p(x)$. This implies $\mathbb{E}f_{x,r} \geq p$. Then by equation 3, we see that $\mathbb{E}_n f \geq \frac{p}{2}$. This implies that all $k_n$ nearest neighbors of $x$ are in the ball $B(x,r)$, and that consequently $\sum_1^n w_i^S(x) 1_{\rho(x,x_i)>r} = 0$. Because this holds for all $x, r$ with $x \in \mathbb{R}^d$ and $r > r_p(x)$, it follows that equation 2 implies that

$$\sup_{x \in X} \sum_1^n w_i^S(x) 1_{\rho(x,x_i)>r_p(x)} = 0.$$

Because equation 3 holds with probability at least $1 - \delta$, and $\delta$ can be made arbitrarily small, the desired claim follows. $\qquad \square$

Let $t_n = \sqrt{dk_n \log n}$.

**Lemma 33.** $\lim_{n \to \infty} E_{S \sim D^n}[t_n \sup_{x \in \mathbb{R}^d} w_i^S(x)] = 0$.

*Proof.* Let $S \sim \mathcal{D}^n$. By the definition of $k_n$ nearest neighbors, $\sup_{x \in \mathbb{R}^d} w_i^S(x) = \frac{1}{k_n}$. Therefore, $t_n \sup_{x \in \mathbb{R}^d} w_i^S(x) = \sqrt{\frac{d \log n}{k_n}}$. By assumption 2. of corollary 12, $\lim_{n \to \infty} \frac{d \log n}{k_n} = 0$, which implies that

$$\lim_{n \to \infty} \mathbb{E}_{S \sim D^n}[t_n \sup_{x \in \mathbb{R}^d} w_i^S(x)] = \lim_{n \to \infty} \sqrt{\frac{d \log n}{k_n}} = \lim_{n \to \infty} \frac{d \log n}{k_n} = 0,$$

as desired. $\qquad \square$

**Lemma 34.** $\lim_{n \to \infty} E_{S \sim D^n} \frac{\log T(W,S)}{t_n} = 0$.

*Proof.* For $S \sim \mathcal{D}^n$, recall that $T(W,S)$ was defined as

$$T(W,S) | \{W_{x,\alpha,\beta} : x \in \mathbb{R}^d, 0 \leq \alpha, 0 \leq \beta \leq 1\}|,$$

where $W_{x,\alpha,\beta}$ denotes

$$W_{x,\alpha,\beta} = \{i : \rho(x,x_i) \leq \alpha, w_i^S(x) \geq \beta\}.$$

Our goal will to be upper bound $\log T(W,S)$.

To do so, we first need a tie-breaking mechanism for $k_n$-nearest neighbors. For each $x_i \in S$, we independently sample $z_i \in [0,1]$ from the uniform distribution. We then tie break based upon the value of $z_i$, i.e. if $\rho(x,x_i) = \rho(x,x_j)$, we say that $x_i$ is closer to $x$ than $x_j$ if $z_i < z_j$. With probability 1, no two values $z_i, z_j$ will be equal, so this ensures that this method always works.

Let $A_{x,\alpha} = \{i : \rho(x,x_i) \leq \alpha\}$ and let $B_{x,c} = \{i : z_i \leq c\}$. The key observation is that for any $\alpha, \beta$, $W_{x,\alpha,\beta} = A_{x,\alpha} \cap B_{x,c}$ for some value of $c$. This can be seen by noting that the nearest neighbors of $x$ are uniquely determined by $\rho(x,x_i)$ and $z_i$. Therefore, it suffices to bound $|A = A_{x,\alpha} : x \in \mathbb{R}^d, \alpha \geq 0\}|$ and $|B = \{B_{x,c} : x \in \mathbb{R}^d, c \geq 0\}|$.

To bound $|A|$, observe that the set of closed balls in $\mathbb{R}^d$ has VC-dimension at most $d + 2$. Thus by Sauer's lemma, there are at most $O(n^{d+2})$ subsets of $\{x_1, x_2, \ldots, x_n\}$ that can be obtained from closed balls. Thus $|A| \leq O(n^{d+2})$.

To bound $|B|$, we simply note that $B_{x,c}$ consists of all $i$ for which $z_i \leq c$. Since the $z_i$ can be sorted, there are at most $n + 1$ such sets. Thus $|B| \leq n + 1$.

Combining this, we see that $T(W, S) \leq |A||B| \leq O(n^{d+3})$. Finally, we see that

$$\lim_{n \to \infty} \frac{\log T(W, S)}{t_n} = \lim_{n \to \infty} \frac{O(d \log n)}{\sqrt{k_n d \log n}} = \lim_{n \to \infty} \sqrt{\frac{O(d \log n)}{k_n}} = 0,$$

with the last inequality holding by condition 2. of Corollary 12.

$\square$

Finally, we note that Corollary 12 is an immediate consequence of the previous 4 lemmas as we can simply apply Theorem 11.

### B.5 Proof of Corollary 13

Let $W$ be a kernel classifier constructed from $K$ and $h_n$ such that the conditions of Corollary 13 hold: that is,

1. $K : [0, \infty) \to [0, \infty)$ is decreasing and satisfies $\int_{\mathbb{R}^d} K(x) dx < \infty$.

2. $\lim_{n \to \infty} h_n = 0$ and $\lim_{n \to \infty} n h_n^d = \infty$.

3. For any $c > 1$, $\lim_{x \to \infty} \frac{K(cx)}{K(x)} = 0$.

4. For any $x \geq 0$, $\lim_{n \to \infty} \frac{n}{\log n} K(\frac{x}{h_n}) = \infty$.

It suffices to show that the conditions of Theorem 11 are met for $W$. Before doing this, we will describe one additional assumption we make for this case.

**Additional Assumption:** We assume that $\mathcal{D}, \mathcal{U}$ are such that there exists some compact set $\mathcal{X} \subset \mathbb{R}^d$ such that for all $x \in supp(\mu)$, $U_x \subset \mathcal{X}$. This is primarily for convenience: observe that any distribution can be approximated arbitrarily closely by distributions satisfying these properties (as each $U_x$ is bounded by assumption). Importantly, because of this, we will note that it is possible for conditions 2. and 3. of Theorem 11 to be relaxed to taking supremums over $\mathcal{X}$ rather than $\mathbb{R}^d$. This is because in our proof, we only ever used these conditions in their restriction to $\bigcup_{x \in supp(\mu)} \bigcup x' \in U_x B(x', r_p(x'))$.

Using this assumption, we return to proving the corollary.

**Lemma 35.** *$W$ is consistent with respect to $\mathcal{D}$.*

*Proof.* Condition 1. of Corollary 13 imply that $K$ is a regular kernel. This together with Condition 2. implies that $W$ is consistent: a proof can be found in [11]. $\square$

To verify the second condition, it will be useful to have the following definition.

**Definition 36.** *For any $p, \epsilon > 0$ and $x \in \mathcal{X}$, define $r_p^\epsilon$ as*

$$r_p^\epsilon(x) = \sup\{r : \mu(B(x, r)) - \mu(B(x, r_p(x))) \leq \epsilon\}.$$

**Lemma 37.** *For any $p, \epsilon > 0$, there exists a constant $c_p^\epsilon > 1$ such that $\frac{r_p^\epsilon(x)}{r_p(x)} \geq c_p^\epsilon$ for all $x \in \mathcal{X}$, where we set $\frac{r_p^\epsilon(x)}{r_p(x)} = \infty$ if $r_p(x) = 0$.*

*Proof.* The basic idea is to use the fact that $\mathcal{X}$ is compact. Our strategy will be to analyze the behavior of $\frac{r_p^\epsilon(x)}{r_p(x)}$ over small balls $B(x_0, r)$ centered around some fixed $x_0$, and then use compactness to pick some finite set of balls $B(x_0, r)$. This must be done carefully because the function $x \to \frac{r_p^\epsilon(x)}{r_p(x)}$ is not necessarily continuous.

Fix any $x_0 \in \mathcal{X}$. First, observe that $r_p^\epsilon(x_0) > r_p(x_0)$. This is because $B(x_0, r_p(x_0)) = \cap_{r > r_p(x_0)} B(x_0, r)$, and consequently $\lim_{r \downarrow r_p(x_0)} \mu(B(x_0, r)) = \mu(B(x_0, r_p(x)))$.

Next, define

$$s_p^\epsilon(x) = \inf\{r : \mu(B(x, r_p(x))) - \mu(B(x, r)) \leq \epsilon\}.$$

We can similarly show that $r_p(x_0) > s_p^\epsilon(x_0)$.

Finally, define

$$r_0 = \frac{1}{3}\min(r_p^\epsilon(x_0) - r_p(x_0), r_p(x_0) - s_p^\epsilon(x_0)).$$

Consider any $x \in B^o(x_0, r_0)$ where $B^o$ denotes the open ball, and let $\alpha = \rho(x_0, x)$. Then we have the following.

1. $r_p(x) \le r_p(x_0) + \alpha$. This holds because $B(x, r_p(x_0) + \alpha)$ contains $B(x_0, r_p(x_0))$, which has probability mass at least $p$.

2. $r_p(x) \ge r_p(x_0) - \alpha$. This holds because if $r_p(x) < r_p(x_0) - \alpha$, then there would exists $r < r_p(x_0)$ such that $\mu(B(x_0, r)) \ge p$ which is a contradiction.

3. $B(x_0, s_p^\epsilon(x_0)) \subset B(x, r_p(x))$. This is just a consequence of the definition of $r_0$ and the previous observation.

By the definitions of $r_p^\epsilon$ and $s_p^\epsilon$, we see that $\mu(B(x_0, r_p^\epsilon(x_0))) - \mu(B(x_0, s_p^\epsilon(x_0))) \le 2\epsilon$. By the triangle inequality, $B(x, r_p^\epsilon(x_0) - \alpha) \subset B(x_0, r_p^\epsilon(x_0))$ and $B(x_0, s_p^\epsilon(x_0)) \subset B(x, r_p(x))$. it follows that

$$\mu(B(x, r_p^\epsilon(x_0) - \alpha)) - \mu(B(x, r_p(x))) \le 2\epsilon,$$

which implies that $r_p^{2\epsilon}(x) \ge r_p^\epsilon(x_0) - \alpha$. Therefore we have the for all $x \in B(x_0, r_0)$,

$$\frac{r_p^{2\epsilon}(x)}{r_p(x)} \ge \frac{r_p^\epsilon(x_0) - \alpha}{r_p(x_0) + \alpha} \ge \frac{2r_p^\epsilon(x_0) + r_p(x_0)}{r_p^\epsilon(x_0) + 2r_p(x_0)}.$$

Notice that the last expression is a constant that depends only on $x_0$, and moreover, since $r_p^\epsilon(x_0) > r_p(x_0)$, this constant is strictly larger than 1. Let us denote this as $c(x_0)$. Then we see that $\frac{r_p^{2\epsilon}(x)}{r_p(x)} \ge c(x_0)$ for all $x \in B^o(x_0, r_0)$.

Finally, observe that $\{B^o(x_0, r_0) : x_0 \in \mathcal{X}\}$ forms an open cover of $\mathcal{X}$ and therefore has a finite sub-cover $C$. Therefore, taking $c = \min_{B^o(x_0, r_0) \in C} c(x_0)$, we see that $\frac{r_p^{2\epsilon}(x)}{r_p(x)} \ge c > 1$ for all $x \in \mathcal{X}$. Because $\epsilon$ was arbitrary, the claim holds. $\qquad\square$

**Lemma 38.** *For any* $0 < p < 1$, $\lim_{n\to\infty} \mathbb{E}_{S\sim\mathcal{D}^n}[\sup_{x\in\mathcal{X}} \sum_1^n w_i^S(x) 1_{\rho(x,x_i)>r_p(x)}] = 0$.

*Proof.* Fix $p > 0$, and fix any $\epsilon, \delta > 0$. Pick $n$ sufficiently large so that the following hold.

1. Let $c_p^\epsilon$ be as defined from Lemma 37.

$$\sup_{x\in\mathcal{X}} \frac{K(c_p^\epsilon r_p(x)/h_n)}{K(r_p(x)/h_n)} < \delta. \tag{4}$$

This is possible because of conditions 2. and 3. of Corollary 13, and because the function $x \to r_p(x)$ is continuous.

2. With probability at least $1 - \delta$ over $S \sim \mathcal{D}^n$, for all $r > 0$, and $x \in \mathcal{X}$,

$$\left|\mu(B(x, r)) - \frac{1}{n}\sum_1^n 1_{x_i \in B(x,r)}\right| \le \epsilon. \tag{5}$$

This is possible because the set of balls $B(x, r)$ has VC dimension at most $d + 2$.

We now bound $\mathbb{E}_{S\sim\mathcal{D}^n}[\sup_{x\in\mathcal{X}} \sum_1^n w_i^S(x) 1_{\rho(x,x_i)>r_p(x)}]$ by dividing into cases where $S$ satisfies and doesn't satisfy equation 5.

Suppose $S$ satisfies equation 5. By condition 1. of Corollary 13, $K$ is decreasing, and by Lemma 37, $r_p^\epsilon(x) \ge c_p^\epsilon r_p(x)$. Therefore, we have that for any $x \in \mathcal{X}$,

$$\sum_1^n K(\rho(x, x_i)/h_n) 1_{\rho(x,x_i)\ge r_p^\epsilon(x)} \le \sum_1^n K(c_p^\epsilon r_p(x)/h_n)$$

$$\le n\delta K(r_p(x)/h_n)),$$

where the second inequality comes from equation 4.

Next, by the definition of $r_p^\epsilon(x)$, we have that $\mu(B(x, r_p^\epsilon(x)) - \mu(B(x, r_p(x)))) \leq \epsilon$. Therefore, by applying equation 5 two times, we see that for any $x \in \mathcal{X}$

$$\sum_1^n K(\rho(x, x_i)/h_n) 1_{r_p(x) < \rho(x,x_i) \leq r_p^\epsilon(x)} \leq 3n\epsilon K(r_p(x)/h_n).$$

Finally, we have that

$$\sum_1^n w_i^S(x) \geq \sum_1^n K(r_p(x)/h_n) 1_{\rho(x,x_i) \leq r_p(x)} \geq n(p-\epsilon)K(r_p(x)/h_n).$$

Therefore, using all three of our inequalities, we have that for any $x \in \mathcal{X}$

$$\sum_1^n w_i^S(x) 1_{\rho(x,x_i) > r_p(x)} = \sum_1^n w_i^S(x) 1_{\rho(x,x_i) > r_p^\epsilon(x)} + \sum_1^n w_i^S(x) 1_{r_p^\epsilon \geq \rho(x,x_i) > r_p(x)}$$

$$= \frac{\sum_1^n K(\rho(x,x_i)/h_n) 1_{\rho(x,x_i) > r_p^\epsilon(x)} + \sum_1^n K(\rho(x,x_i)/h_n) 1_{r_p^\epsilon \geq \rho(x,x_i) > r_p(x)}}{\sum_1^n K(\rho(x,x_i)/h_n)}$$

$$\leq \frac{n\delta K(r_p(x)/h_n)) + 3n\epsilon K(r_p(x)/h_n)}{n(p-\epsilon)K(r_p(x)/h_n)}.$$

$$= \frac{\delta + 3\epsilon}{p - \epsilon}.$$

If $S$ does *not* satisfy equation 5, then we simply have $\sup_{x \in \mathcal{X}} \sum_1^n w_i^S(x) 1_{\rho(x,x_i) > r_p(x)} \leq 1$. Combining all of this, we have that

$$E_{S \sim \mathcal{D}^n} \sum_1^n w_i^S(x) 1_{\rho(x,x_i) > r_p(x)} \leq \delta(1) + (1 - \delta) \frac{\delta + 3\epsilon}{p - \epsilon}.$$

Since $\delta, \epsilon$ can be made arbitrarily small, the result follows. $\qquad \square$

By assumption, $\mathcal{X}$ is compact and therefore has diameter $D < \infty$. Define

$$t_n = \sqrt{n \log n K(\frac{D}{h_n})} \text{ for } 1 \leq n < \infty.$$

**Lemma 39.** $\lim_{n \to \infty} E_{S \sim D^n}[t_n \sup_{x \in \mathcal{X}} w_i^S(x)] = 0$.

*Proof.* Because $K$ is a decreasing function, we have that $K(D/h_n) \leq K(\rho(x, x_i)/h_n) \leq K(0)$. As a result, we have that for any $x \in \mathcal{X}$,

$$t_n \sup_{1 \leq i \leq n} w_i^S(x) = \frac{t_n \sup_{1 \leq i \leq n} K(\rho(x, x_i)/h_n)}{\sum_1^n K(\rho(x, x_i)/h_n)}$$

$$\leq \frac{t_n K(0)}{nK(D/h_n)}$$

$$= K(0)\sqrt{\frac{n \log n K(D/h_n)}{n^2 K(D/h_n)^2}}$$

$$= K(0)\sqrt{\frac{\log n}{nK(D/h_n)}}.$$

However, by condition 4. of Corollary 13, $\lim_{n \to \infty} \frac{n}{\log n} K(D/h_n) = \infty$. Therefore, since the above inequality holds for all $x \in \mathcal{X}$, we have that

$$\lim_{n \to \infty} E_{S \sim D^n}[t_n \sup_{x \in \mathcal{X}} w_i^S(x)] \leq \lim_{n \to \infty} K(0)\sqrt{\frac{\log n}{nK(D/h_n)}} = 0.$$

$\qquad \square$

**Lemma 40.** $\lim_{n \to \infty} E_{S \sim \mathcal{D}^n} \frac{\log T(W,S)}{t_n} = 0.$

*Proof.* For $S \sim \mathcal{D}^n$, recall that $T(W,S)$ was defined as

$$T(W,S)|\{W_{x,\alpha,\beta} : x \in \mathcal{X}, 0 \le \alpha, 0 \le \beta \le 1\}|,$$

where $W_{x,\alpha,\beta}$ denotes

$$W_{x,\alpha,\beta} = \{i : \rho(x, x_i) \le \alpha, w_i^S(x) \ge \beta\}.$$

Our goal will to be upper bound $\log T(W,S)$.

The key observation is that $W_{x,\alpha,\beta}$ is precisely the set of $x_i$ for which $\rho(x, x_i) \le r$ where $r$ is some threshold. This is because the restriction that $w_i^S(x) \ge \beta$ can be directly translated into $\rho(x, x_i) \le r$ for some value of $r$, as $K$ is a monotonically decreasing function. Thus, $T(W,S)$ is the number of subsets of $S$ that can be obtained by considering the interior of some ball $B(x,r)$ centered at $x$ with radius $r$.

We now observe that the set of closed balls in $\mathbb{R}^d$ has VC-dimension at most $d + 2$. Thus by Sauer's lemma, there are at most $O(n^{d+2})$ subsets of $\{x_1, x_2, \ldots, x_n\}$ that can be obtained from closed balls. Thus $T(W,S) \le O(n^{d+2})$.

Finally, we see that

$$\lim_{n \to \infty} \frac{\log T(W,S)}{t_n} = \lim_{n \to \infty} \frac{O(d \log n)}{\sqrt{n \log n K(\frac{D}{h_n})}} \le \lim_{n \to \infty} \sqrt{\frac{O(d \log n)}{n K(\frac{D}{h_n})}} = 0,$$

with the last equality holding by condition 4. of Corollary 13. $\qquad\square$

Finally, we note that Corollary 13 is an immediate consequences of Lemmas 35, 38, 39, and 40, as we can simply apply Theorem 11.

## C  Useful Technical Definitions and Lemmas

**Lemma 41.** *Let $\mu$ be a measure over $\mathbb{R}^d$, and let $\mathcal{A}$ denote a countable collections of measurable sets $A_i$ such that $\mu(\bigcup_{A \in \mathcal{A}} A) < \infty$. Then for all $\epsilon > 0$, there exists a finite subset of $\mathcal{A}$, $\{A_1, \ldots, A_m\}$ such that*

$$\mu(A_1 \cup A_2 \cup \cdots \cup A_m) > \mu(\bigcup_{A \in \mathcal{A}} A) - \epsilon.$$

*Proof.* Follows directly from the definition of a measure. $\qquad\square$

### C.1  The support of a distribution

Let $\mu$ be a probability measure over $\mathbb{R}^d$.

**Definition 42.** *The **support** of $\mu$, $supp(\mu)$, is defined as all $x \in \mathbb{R}^d$ such that for all $r > 0$, $\mu(B(x,r)) > 0$.*

From this definition, we can show that $supp(\mu)$ is closed.

**Lemma 43.** *$supp(\mu)$ is closed.*

*Proof.* Let $x$ be a point such that $B(x,r) \cap supp(\mu) \ne \emptyset$ for all $r > 0$. It suffices to show that $x \in supp(\mu)$, as this will imply closure.

Let $x$ be such a point, and fix $r > 0$. Then there exists $x' \in B(x, r/2)$ such that $x' \in supp(\mu)$. By definition, we see that $\mu(B(x', r/3)) > 0$. However, $B(x', r/3) \subset B(x, r)$ by the triangle inequality. it follows that $\mu(B(x,r)) > 0$. Since $r$ was arbitrary, it follows that $x \in supp(\mu)$. $\qquad\square$

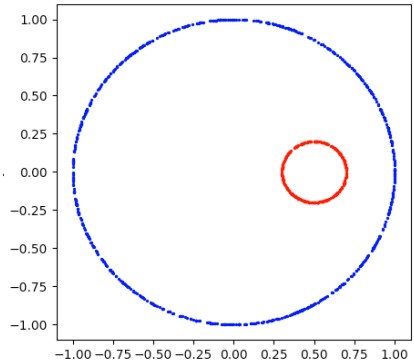

Figure 5: Our data distribution $\mathcal{D} = (\mu, \eta)$ with $\mu^+$ shown in blue and $\mu^-$ shown in red. Observe that this simple distribution captures varying distances between the red and blue regions, which necessitates having varying sizes for robustness regions.

## D   Experiment Details

**Data Distribution**   Our data distribution $\mathcal{D} = (\mu, \eta)$ is over $\mathbb{R}^2 \times \{\pm 1\}$, and is defined as follows. We let $\mu^+$ consist of a uniform distribution over the circle $x^2 + y^2 = 1$, and $\mu^-$ consist of the uniform distribution over the circle $(x - 0.5)^2 + y^2 = 0.04$. The two distributions are weighted so that we draw a point from $\mu^+$ with probability 0.7, and $\mu^-$ with probability 0.3. Finally, we utilize label noise 0.2 meaning that the label $y$ matches that given by the Bayes optimal with probability 0.2. In summary, $\mathcal{D}$ can be described with the following 4 cases:

1. With probability $0.7 \times 0.8$, we select $(x, y)$ with $x \in \mu^+$ and $y = +1$.

2. With probability $0.7 \times 0.2$, we select $(x, y)$ with $x \in \mu^+$ and $y = -1$.

3. With probability $0.3 \times 0.8$, we select $(x, y)$ with $x \in \mu^-$ and $y = -1$.

4. With probability $0.3 \times 0.2$, we select $(x, y)$ with $x \in \mu^-$ and $y = +1$.

We also include a drawing (Figure 5) of the support of $\mathcal{D}$, with the positive portion $\mu^+$ shown in blue and the negative portion, $\mu^-$ shown in red.

**Computing Robustness Regions**   Recall that in order to measure robustness, we utilize the so-called partial neighborhood preserving regions $V_x^\kappa$ (Definition 6) for varying values of $\kappa$. In the case of our data distribution $\mathcal{D}$, $V_x^\kappa$ consists of points closer to $x$ by a factor of $\kappa$ than they are to $\mu^-$ (resp. $\mu^+$) when $x \in \mu^+$ (resp. $\mu^-$). To represent a region $V_x^\kappa$, we simply use a function $f$ that verifies whether a given point $x' \in V_x^\kappa$. While this methodology is not sufficient for training general classifiers (for a whole litany of reasons: to begin with it assumes full knowledge of the distribution), it will suffice for our toy synthetic experiments.

**Trained Classifiers**   We train two classifiers, both of which are kernel classifiers.

The first classifier is an exponential kernel classifier with bandwidth function $h_n = \frac{1}{10\sqrt{\log n}}$ and kernel function $K(x) = e^{-x}$.

The second classifier is a polynomial kernel classifier with bandwidth function $h_n = \frac{1}{10n^{1/3}}$ and kernel function $K(x) = \frac{1}{1+x^2}$.

Both of these kernels are regular kernels, and both bandwidths satisfy sufficient conditions for consistency with respect to accuracy. In other words, both of these classifiers will converge towards the accuracy of the Bayes optimal.

However, the first classifier is selected to satisfy the criterion of Corollary 13, whereas the second is not. This distinction is reflected in our experiments.

**Verifying Robustness** To verify the robustness of classifier $f$ at point $x$ (with respect to $V_x^\kappa$), we simply do a grid search with grid parameter 0.01. We grid the entire regions into points with distance at most 0.01 between them, and then verify that $f$ has the desired value at all of those points. To ensure proper robustness, we also simply verify that $f$ cannot change enough within a distance of 0.01 by constructing an upper bound on how much $f$ can possibly change. For kernel classifiers, this is simple to do as there is a relatively straightforward upper bound on the gradient of a Kernel classifier.