# OpenReview forum: "Consistent Non-Parametric Methods for Maximizing Robustness"
_NeurIPS.cc/2021/Conference — NeurIPS 2021 Poster_

### Official Review · Reviewer_7R4w · 2021-07-14

**Rating:** 7
**Confidence:** 3

**Summary:**

The authors argue that robustness regions should be larger in some regions of data, and smaller in others, and propose a new limit classifier, called the neighborhood optimal classifier, that extends the Bayes optimal classifier  outside its support by using the label of the closest in-support point. They argue that this classifier maximizes the size of its robustness regions subject to the constraint of having accuracy equal to the Bayes optimal, then present sufficient conditions under which general non-parametric methods that can be represented as weight functions converge towards this limit object, and show that both nearest neighbors and kernel classifiers (under certain assumptions) suffice.

**Ethical Concerns:**

No.

**Limitations And Societal Impact:**

The authors did not address the limitations and potential negative societal impact of their work.  A paragraph of discussion is needed.

**Main Review:**

The authors propose an alternative formulation of robust classification that ensures that in the large sample limit, there is no robustness-accuracy tradeoff, and that regions of space with higher separation are classified more robustly. To this end, the authors introduce a new large-sample limit, the neighborhood preserving Bayes optimal (NPBO) classifier. For an input x, it outputs the prediction of the Bayes Optimal on the nearest neighbor (including itself) in the support of the data distribution D.


They show that k_n-nearest neighbors converge to the NPBO provided k_n = ω(log n), and kernel classifiers converge to the NPBO provided certain conditions (the kernel function K has faster than polynomial decay, and the bandwidth parameter h_n decreases sufficiently slowly), but certain types of histogram classifiers do not converge to the NPBO, even if they do converge to the Bayes optimal. They also provide experiments to validate their claims.

I feel this work is novel in motivation and solid in theory.

I have the following comments for improving the writing and organization:
In Section 1, "astuteness" is used without any citation in the first paragraph.
In Section 2, there are many conceptions that are existed in previous papers, but the authors did not mention or give references,
for instance, Definition 1 "astute and astuteness", weight functions, histogram classfiers, etc.
Line 259: Conditions 2. and 3. are ..., conditions 2. and 3. of ... .  I think the period "." is not needed after 2 and 3.
Line 144: We call this the the neighborhood preserving ...





**Time Spent Reviewing:**

20 hours

---

> ### Author Response · Authors · 2021-08-10
> **Response to reviewer 7R4w**
>
> Thank you for your feedback. We will definitely include citations for many of the common terms used in this area -- apologies for missing those.
>
> The main limitation of this work is that the results hold in the large-n regime. In terms of societal consequences, this is a largely theoretical paper which does not appear to directly have any negative societal impact. This work takes a first step towards a data-driven formulation of robustness that does not hamper accuracy. A distant impact of this work is in designing classifiers that balance robustness and accuracy more effectively; this could have a negative impact if these robust classifiers are used to cause harm. We will add a paragraph to this effect.

---

### Official Review · Reviewer_piq5 · 2021-07-16

**Rating:** 5
**Confidence:** 3

**Summary:**

This paper aims to give a new robust learning framework where the radius of adversarial attacks at different points can adaptively change. The main purpose is "to increase robustness, without sacrificing accuracy", as claimed by the author(s). In order to achieve this goal, author(s) have utilized a number of consistent non-parametric learning algorithms which are already proved to converge to an *optimal* classifier as the sample size increases (optimality could be defined in any certain sense, however, in this paper Bayes optimality has been mostly focused). Then, a new modification has been proposed in order to help such learning techniques to preserve as much adversarial robustness as they can, while still converge to the Bayes optimal classifier.

The main results are (most probably) solid, and maybe even interesting specially when we look at them from a pure theoretical standpoint. Moreover, I checked all the theoretical results and some of their proofs, and didn't find any notable technical mistakes or counter-intuitive conclusions. However, paper suffers from uninformative writing (specially in the introduction section) and lack of proper justification for some of the core ideas in real-world applications. The above problems, in addition to the fact that results are mostly asymptotic and centered around "consistency" of classifiers rather than their finite-sample performance, are the main weaknesses of the paper.

My vote at the current stage is weak-reject.

**Limitations And Societal Impact:**

No problem here.

**Main Review:**

As far as I have understood, the main contribution of this paper is as follows: When the number of samples $n$ goes toward infinity and consequently the true data distribution becomes known to the learner,  then "the most robust classifier which coincides with the Bayes optimal classifier inside the support" would be trivial and easy to achieve. This paper gives a methodical approach so that some classes of non-paramteric classifiers (like $k$-nearest neighbor) can be computed for finite $n$, while are also guaranteed to converge to the trivial robust classifiers of $n=\infty$ which are described above (The guarantees are all asymptotic in this work). There are some other contributions in proving the consistency of certain non-parametric classes of learners, which I assume to be additional results.

---------------------------------------------------------------------------

Main comments:

Paper can greatly benefit from an improved writing, specially for the Introduction section. Some terms, such as "accuracy and astuteness" have been carelessly used in this section without being properly defined (they are both defined later in subsequent sections). This has made the Intro part to be somehow uninformative. Also, some motivations in the beginning of the paper have become lost and buried under tones of theorems and definitions inside the manuscript. For example, the abstract of the paper gives some motivations w.r.t. varying radius of attacks $r$ for different regions of the distribution. However, I do not see the results to be that much aligned with this particular motif. Maybe the abstract and parts of the introduction section should be rewritten.

Another issue is that paper lacks real-world justification for its main idea: Having robustness to out-of-distribution samples (for example, off-manifold data points which are usually the result of adversarial attacks) is interesting, but the discussions around this issue in the paper is slim to none. In other words, why should any reader be concerned with samples like $x$ which are not in the support of data distribution, i.e. $x\notin\mathrm{supp}\left(\mu\right)$? One might assume the answer to be crystal clear for some well-informed audience, but that's not the case for general readers.

Another fundamental problem is that results in this work are asymptotic. No non-asymptotic bounds have been proposed to guarantee any type of performance measure when the number of samples $n$ is finite. Any type of guarantees on the adversarial robustness when $n$ is infinite and the true underlying data distribution (or at least the true data manifold) is revealed are not that much interesting. It would be nice if authors can give some certificate for robustness when $n$ is finite. Some of the cited works in this paper (Sinha et al., for example) have already done that for distributionally robust scenarios.

--------------------------------------------------

Minor comments:
-(Line 144): "the" has been repeated.

**Time Spent Reviewing:**

5 hours

---

> ### Author Response · Authors · 2021-08-10
> **response to reviewer piq5**
>
> Thank you for your feedback. It seems that the main concerns are the asymptotic nature of our results and exposition.
>
> For this paper, we chose to focus on asymptotics because of generality -- Stone’s theorem provides a general framework which we can extend to prove asymptotic convergence for a large number of non-parametric classifiers. In contrast, finite sample results on accuracy are only known on a case-by-case basis -- eg, there are different techniques for analyzing nearest neighbors, decision trees, and so on. We agree that is a very interesting direction of future work, and are actively working on it.
>
> Thank you for your suggestions on the writing. We are happy to make the introduction more accessible to a general audience. Regarding varying balls of radius $r$, what we intended to communicate was that our notion of neighborhood preserving robustness regions generalizes the ideas of varying the radii. These regions are tailored to every point (in both size and shape) based on the support of the distribution.

---

> > ### Comment · Reviewer_piq5 · 2021-08-31
> > **Final comment**
> >
> > Thank you for your response, and sorry for my late reply,
> >
> > I went through other reviews and also read author(s)' response. I feel that my main concerns are not answered yet. However, given that two other reviewers are very positive with respect to this paper, I also give the possibility that I might have missed the significance of this contribution. Therefore, I keep my score but reduce my confidence, so other reviewers could have higher weights for their vote.

---

### Official Review · Reviewer_n6N3 · 2021-07-16

**Rating:** 7
**Confidence:** 3

**Summary:**

This paper studies classification rules that are "astute", that is, those that are accurate and robust. Importantly, the authors note that common definitions of robust classifiers, which are given in terms of a fixed parameter (typically a size of a perturbation set) are not astute, as they do not fully exploit the fact that margins are not uniform across the data distribution. This paper first studies conditions under their so-called "neighborhood optimal classifier" maximizes robustness while retaining the Bayes classifier accuracy in the unperturbed distribution. Furthermore, the authors show that several popular non-parametric classifier provide consistent estimates for this classifier. Finally, this is demonstrated numerically in a simple experimental setting.

**Limitations And Societal Impact:**

Adequately addressed.

**Main Review:**

This paper is very interesting, and a pleasure to read. The limitation of current approaches to study adversarial robustness is clear from the presented concepts, and the proposed solutions are intuitive and elegant. I have a few minor comments:

Main comments:
- While the authors present results of consistency of certain non-parametric classifiers, their finite-sample performance is only evaluated numerically. Can the authors comment, at least briefly, on the potential of extending the analysis of convergence rates of non-parametric classifiers (such as those in [Doring et al, 2018]) to the setting presented here?

- In the paragraph on line 154, the authors motivate their definition of "neighborhood preserving robustness regions". To do so, they explain that they seek for regions U_x so that the astute accuracy of g_max equals the accuracy of the Bayes classifier. However, it's not totally clear what g_max is. Is it the one that maximizes the measure of U_x? A clarification would be appreciated.

- On the paragraph following Theorem 5, the authors comment that g_neighbor can be thought of a local maximum to a constrained optimization problem. However, it is not clear why this is not a global maximum - how could one obtain a classifier with larger robustness constrained to having equal accuracy as the Bayes?

Minor comments:
- Lines 57: space missing on "Instead [3]"
- The paragraphs on line 71 and 78 feel a bit repetitive, given that the preceding paragraphs just gave a summary of the presented contributions.
- Line 96: coma instead of semicolon?
- Line 137: principal -> principle
- Line 144: "the the"
- Line 171: the authors write "max-margin Bayes" classifier, but I believe they mean the neighborhood preserving Bayes classifier? This might be equivalent, but has not been defined as "max-margin" explicitely.
- The notation of algorithm (A_S) and Accuracy is slightly confusing.
- Line 222 "consistnecy" -> consistency
- Line 222: "needed accuracy" -> "needed for accuracy"
- On Corollary 12, the authors use the notation k_n^\infty_1. There's a parenthesis that seems out of place. But also, this notation has not been defined?
- On Corollary 13, the authors use h_n, but this seems undefined. Do they simply refer to the rule k_nn on the computed features?
- Line 293: "The answer this" -> the answer to this
- Figure 3 seems to have some strange gray edges on its bottom and left sides.
- Line 335: "performs will" -> performs well.


**Time Spent Reviewing:**

2

---

> ### Author Response · Authors · 2021-08-10
> **Response to reviewer n6N3**
>
> Thank you for your feedback and writing suggestions.
>
> 1. We agree that extending our analysis to convergence rates is a very interesting and important next step for our work. While it might be tricky to find a general result for rates (especially with regards to weight classifiers), we believe that it is possible to obtain some sort of result for nearest neighbors in particular. Our framework seems somewhat amenable to utilizing the analysis style given by Chaudhuri and Dasgupta (NIPS 2014), and this is a direction we are currently pursuing.
>
> 2. We apologize for this poorly used notation: we never did define $g_{max}$. What we meant to communicate was that we desire regions such that $\max_g A_{\mathcal{U}}(g, \mathcal{D}) = A(g_{bayes}, \mathcal{D}).$ Thus, “$g_{max}$” is supposed to represent the maximally astute classifier with respect to a given set of regions $\mathcal{U}$ (which is more or less what you surmised).
>
> 3. Theorem 5 shows that no classifier can have __strictly__ better robustness than the neighborhood preserving Bayes optimal classifier. However, it is possible to construct classifiers that improve upon the robustness in some locations. For instance consider a distribution of 2 points, $x = 0, x = 1$ that are equally weighted and oppositely labeled. The neighborhood preserving bayes would have a decision boundary at $x = 0.5$. However, it is possible that perhaps a larger robustness region at $x = 0$ is desired (for some application specific reasons) and consequently a decision boundary at 0.8 might be better in practice. Theorem 5 points out that this necessarily comes at a cost of robustness at $x = 1$. For this reason, we hesitate to call the neighborhood preserving Bayes the global optimum of robustness.

---

### Official Review · Reviewer_z1YM · 2021-07-19

**Rating:** 5
**Confidence:** 3

**Summary:**

This paper argues that the standard robust learning method ignores the data heterogeneity: the robustness radius is fixed as some artificial value r for all the examples. The authors then propose a new neighborhood optimal classifier to address this problem, and theoretically studies the convergence condition for a class of non-parametric methods, including histogram classifiers, nearest neighbors, and kernel classifiers. Synthetic experiments are provided to verify their theoretical results.

**Limitations And Societal Impact:**

Yes

**Main Review:**

The proposed robustness framework is well motivated and clearly defined. At a high level, introducing heterogeneous robustness regions for different examples seems to be a promising direction for a better robustness definition. The theoretical results are also clearly presented and easy to follow. Nevertheless, my main concern of this paper is the practicability of the proposed framework: the proposed definition of neighborhood optimal classifier requires the knowledge of the support of the conditional distribution, which seems to be not practical for typical classification problem. More specifically, I have the following questions:

1.	How to apply Definition 3 to a more typical finite-sample classification problem? Are there ways to estimate the support of the ground-truth conditional distribution?

2.	What does the theoretical results suggest for robust learning using non-parametric methods? How can these results generalize beyond non-parametric methods?

3.	The dataset of overlayed circles considered in Section 5 seems to be too artificial to me. Including the experimental results on other datasets, such as Gaussian mixtures or even some image benchmarks, will strength the paper.


**Time Spent Reviewing:**

2 hours

---

> ### Author Response · Authors · 2021-08-10
> **response to reviewer z1YM**
>
> Thank you for your feedback. It sounds like the main concern is about the practicality of our framework; perhaps we can clarify. We will also explicitly address and clarify this in the final version of our paper.
>
> Just like the Bayes Optimal, computing the neighborhood-optimal and neighborhood preserving robustness regions explicitly requires explicit knowledge of the support. However, we believe that this does not preclude this notion from having practical relevance. This is for two reasons.
>
> First, our results show that it is possible to construct algorithms that converge towards robust classifiers in our framework despite not having explicit knowledge of these regions. This is the primary practical result of our paper: it gives general criteria for designing robust non-parametric classifiers within our framework.
>
> Second, the use of abstract support dependence notions is useful for understanding how well algorithms converge towards desirable criteria. As an example, knowledge of the Bayes Optimal classifier also requires full knowledge of the data’s support (which isn’t achievable in the finite sample limit) but nevertheless is still useful as an abstraction for understanding non-parametric classifiers. We view our framework as similar: while we can’t explicitly compute the neighborhood optimal classifier or the neighborhood preserving robustness regions, they serve as important limit objects for building robust classifiers.
>
> Here are answers to your specific questions.
>
> 1. Definition 3 serves as the “ideal” robustness region with which to evaluate your classifier. For finite-sample problems, this region isn’t precisely computable, but it nevertheless is possible to build classifiers that eventually converge towards this region.
>
> 2. Our results suggest that non-parametric classifiers with broader precision (i.e. larger values of $k_n$ or kernels with more specific kernel functions) have better robustness properties in the large sample limit. This improvement in robustness will hold for any sort of robustness regions that are subsets of the optimal neighborhood preserving robustness regions. While performance in the large sample limit does not guarantee better performance for finite samples, it does suggest a direction towards running experiments and can be useful in a case by case basis.
> Regarding generalization beyond non-parametric methods, this is an active area of further research for us. We agree that in this case, it will be important to develop methods for estimating the “neighborhood robustness” loss for a parametric classifier. Note that this loss estimation was unnecessary for building non-parametric classifiers.
>
> 3. We initially only included simple experiments as the main contribution of our paper is the theory, but we agree that including more examples would be helpful. In a preliminary step towards this direction, we have rerun our experiments for a more standard mixture of 2 gaussian distributions (2 dimensional with equal covariance matrices) with results presented in the following tables. In each table, the first columns is training size, the second test is accuracy, and each subsequent columns is the testing astuteness for a given value of kappa (the parameter determining the "size" of the robustness region). The results in these tables similarly validate our theories: the exponential kernel converges in both astuteness and accuracy (albeit more slowly than in it did for the circle distribution) whereas the polynomial kernel degrades in performance as kappa increases.
>
> **Polynomial Kernel:**
>
> **Training Size**|**acc**|**kappa=0.1**|**kappa=0.3**|**kappa=0.5**
> :-----:|:-----:|:-----:|:-----:|:-----:
> 400|0.84|0.98|0.72|0.48
> 800|0.94|0.82|0.7|0.5
> 1200|0.86|0.84|0.62|0.24
> 1600|0.92|0.9|0.44|0.34
> 2000|0.92|0.76|0.58|0.38
> 2400|0.84|0.94|0.6|0.2
> 2800|0.94|0.82|0.44|0.24
> 3200|0.84|0.82|0.44|0.14
> 3600|0.88|0.8|0.48|0.24
> 4000|0.9|0.84|0.42|0.18
> 5000|0.92|0.8|0.44|0.26
> 7000|0.98|0.84|0.38|0.22
> 10000|0.94|0.84|0.4|0.36
>
> **Exponential Kernel:**
>
> **Training Size**|**acc**|**kappa=0.1**|**kappa=0.3**|**kappa=0.5**
> :-----:|:-----:|:-----:|:-----:|:-----:
> 400|0.92|0.9|0.7|0.4
> 800|0.92|0.88|0.6|0.58
> 1200|0.96|0.84|0.64|0.42
> 1600|0.84|0.84|0.48|0.3
> 2000|0.9|0.82|0.58|0.32
> 2400|0.94|0.82|0.62|0.46
> 2800|0.88|0.94|0.7|0.54
> 3200|0.84|0.86|0.8|0.44
> 3600|0.9|0.84|0.7|0.52
> 4000|0.9|0.88|0.84|0.56
> 5000|0.86|0.92|0.72|0.72
> 7000|0.92|0.92|0.88|0.74
> 10000|0.94|0.92|0.9|0.74

---

> > ### Author Response · Authors · 2021-08-20
> > **Checking on response**
> >
> > Hi, we just wanted to check whether our response addressed your concerns - specifically with regards to finite sample viability and the datasets for the experiments. Please let us know if you have any follow-up clarifications or questions. Thanks!

---

### Author Response · Authors · 2021-08-30
**Comments**

Hello Reviewers!

Thank you again for your detailed and helpful reviews. We are just checking to see if there are any last clarifications or questions that we can respond to before the discussion period ends.

---

### Decision · Program_Chairs · 2021-09-27

**Decision:**

Accept (Poster)

**Comment:**

The paper presents a novel and interesting idea to combine robustness and consistency.
It is well written and provides explicit examples how the general framework can be
applied. On the negative side, only consistency is considered and more refined convergence
results such as learning rates or finite sample bounds are missing.